# Feature Alignment: Rethinking Efficient Active Learning via Proxy in the Context of Pre-trained Models

**Ziting Wen**                                                    *zwen4889@uni.sydney.edu.au*
*Australian Centre for Robotics, University of Sydney*

**Oscar Pizarro**                                                 *oscar.pizarro@ntnu.no*
*Department of Marine Technology, Norwegian University of Science and Technology*
*Australian Centre for Robotics, University of Sydney*

**Stefan Williams**                                               *stefan.williams@sydney.edu.au*
*Australian Centre for Robotics, University of Sydney*

**Reviewed on OpenReview:** *https://openreview.net/forum?id=PNcgJMJcdl*

## Abstract

Fine-tuning the pre-trained model with active learning holds promise for reducing annotation costs. However, this combination introduces significant computational costs, particularly with the growing scale of pre-trained models. Recent research has proposed proxy-based active learning, which pre-computes features to reduce computational costs. Yet, this approach often incurs a significant loss in active learning performance, sometimes outweighing the computational cost savings. This paper demonstrates that not all sample selection differences result in performance degradation. Furthermore, we show that suitable training methods can mitigate the decline of active learning performance caused by certain selection discrepancies. Building upon detailed analysis, we propose a novel method, aligned selection via proxy, which improves proxy-based active learning performance by updating pre-computed features and selecting a proper training method. Extensive experiments validate that our method improves the total cost of efficient active learning while maintaining computational efficiency. The code is available at `https://github.com/ZiTingW/asvp`.

## 1 Introduction

Training effective deep neural networks typically requires large-scale data (Deng et al., 2009; He et al., 2016; Liu et al., 2021). However, the high cost of acquiring data, especially annotated data, poses a significant challenge for practitioners (Budd et al., 2021; Norouzzadeh et al., 2021). Active learning and the pre-training fine-tuning paradigm are widely adopted strategies to address this challenge. Active learning reduces the demand for extensive annotation by iteratively selecting and labeling the most informative samples (Ren et al., 2021). Additionally, the pre-training fine-tuning method leverages large-scale unsupervised pre-training to create a powerful foundational model, enabling exceptional performance on downstream tasks with only a limited amount of labeled data (Chen et al., 2020a; Grill et al., 2020; Caron et al., 2021; He et al., 2022). To further minimize labels' demands, researchers have turned their attention to active fine-tuning, which fine-tunes the pre-training model with labeled samples selected by active learning strategies (Xie et al., 2023; Chan et al., 2021; Bengar et al., 2021). However, as the scale of pre-training models continues to grow, the sample selection time of active learning, which often is overlooked in traditional active learning, becomes a challenge (Coleman et al., 2019). As illustrated in fig. 1, larger pre-training models lead to higher accuracy, but also significantly increase the time required for active learning sample selection.

To address this issue, recent research has introduced an efficient active learning framework known as Selection via Proxy based on pre-trained features **(SVPp)** (Zhang et al., 2024). SVPp begins by forwarding the entire

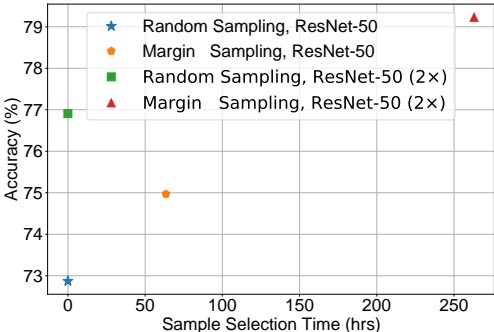
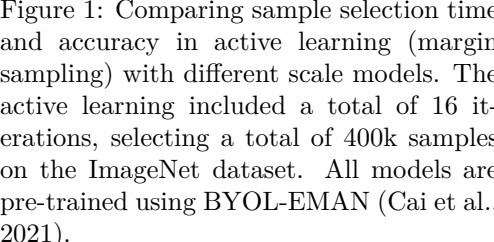
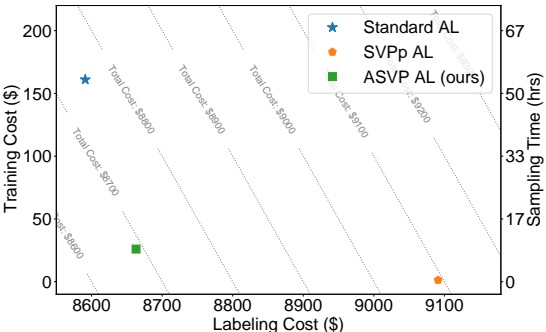

Figure 1: Comparing sample selection time and accuracy in active learning (margin sampling) with different scale models. The active learning included a total of 16 iterations, selecting a total of 400k samples on the ImageNet dataset. All models are pre-trained using BYOL-EMAN (Cai et al., 2021).

Figure 2: Comparison of the labeling and training costs across random sampling, the standard active learning pipeline, the efficient active learning method (SVPp), and our method (ASVP), using margin sampling to achieve the same accuracy as randomly selecting 400k samples on the ImageNet dataset. A ResNet-50 model was employed, with training costs calculated based on AWS EC2 P3 instances (following Zhang et al., 2024), and annotation costs estimated using AWS Mechanical Turk[1] with triple reviews.

dataset through a pre-trained model once, recording pre-trained features for all samples. Subsequently, in multiple iterations of active learning, a simple MLP classifier is trained using the pre-computed features for sample selection. After the sample selection, the entire model is fine-tuned once. However, we observe that, in comparison to the **standard active learning** method (fine-tuning the whole pre-trained model in each active learning iteration), SVPp may compromise the effectiveness of active learning, resulting in additional annotation costs. As depicted in fig. 2, we noticed that while SVPp significantly reduces active learning time (training costs), its decline in active learning performance leads to a notable increase in annotation costs. Practitioners therefore have to face the dilemma of weighing computational efficiency against overall cost (the sum of labeling cost and the training cost).

In light of this challenge, in this paper, we analyze the factors contributing to the decline in active learning performance when using SVPp in sec. 4. Beyond the intuition-aligned reason for SVPp active learning performance decline: fine-tuned features being more capable of distinguishing sample categories than pre-trained features, leading to the selection of redundant samples (i.e., samples the fine-tuned model can already predict correctly), our analysis reveals an intriguing discovery: when pre-trained features are of comparable quality to fine-tuned features, not all sample selection differences result in a decline in SVPp performance. Moreover, some performance drops are due to samples selected by SVPp causing substantial modifications to the pre-trained model during fine-tuning.

Based on these insights, we propose an aligned selection via proxy strategy, **ASVP**, to enhance the performance of efficient active learning in sec. 5. First, we align the pre-computed features used by the proxy model with the one used in standard active learning. Specifically, we update the pre-computed features based on the fine-tuned model when the pre-trained features' ability to distinguish sample categories significantly lags behind the fine-tuned features. In addition, we switch the training method between linear-probing then fine-tuning (LP-FT) (Kumar et al., 2022) and fine-tuning (FT) to mitigate the impact of samples selected by the proxy model on the model's performance.

Extensive experimental validation shows that our method improves the effectiveness of efficient active learning, achieving similar or better total cost savings compared to those of standard active learning, while still maintaining computational efficiency.

---

[1]https://aws.amazon.com/sagemaker/groundtruth/pricing/

The contributions of this paper can be summarized as follows: (1) This paper empirically analyzes which sample selection differences contribute to the performance drop when employing SVPp. (2) We propose a novel efficient active learning approach, aligned selection via proxy (ASVP), based on the detailed analysis. It improves the performance of efficient active learning while incurring a marginal amount of computation time. (3) We introduce a novel and practical evaluation metric for efficient active learning: the sample saving ratio. This metric directly quantifies the savings in annotation achieved by employing active learning strategies compared with the random baseline. It also facilitates the estimation of overall savings, including savings in both computational and labeling costs, thereby assisting practitioners in making a decision on whether to use efficient active learning strategies.

## 2 Related Work

**Active learning** is a technique that minimizes the number of labels needed for model training by selectively requesting annotations for samples. Numerous active learning strategies have been proposed, primarily including uncertainty sampling (Lewis & Catlett, 1994; Scheffer et al., 2001; Gal et al., 2017; Kirsch et al., 2019; Woo, 2022), diversity sampling (feature space coverage) (Sener & Savarese, 2018; Mahmood et al., 2022), their combinations (Yang et al., 2017; Ash et al., 2020; Shui et al., 2020), and some learning-based approaches (Sinha et al., 2019; Yoo & Kweon, 2019; Tran et al., 2019). With the advancement of large-scale unsupervised pre-training and fine-tuning, an increasing number of researchers have turned their attention to active fine-tuning, which fine-tuning the pre-trained model with samples selected by active learning strategies. Given the success of the pre-training fine-tuning mode, most works of active fine-tuning have focused on developing active learning strategies that can work with extremely limited annotations (Hacohen et al., 2022; Yehuda et al., 2022; Wen et al., 2023). Furthermore, researchers have validated the performance of existing active learning methods when applied to pre-training fine-tuning mode, demonstrating that traditional margin sampling exhibits strong performance (Zhang et al., 2024; Emam et al., 2021).

**Efficient active learning.** Most active learning methods operate in a multi-round mode, iteratively training models and selecting samples to label. This mode needs significant training time and costs. In the context of active fine-tuning, this phenomenon becomes more pronounced, as larger models are better equipped to effectively leverage data in unsupervised pre-training (Chen et al., 2020b; Oquab et al., 2023).

Increasing the active learning batch size, which refers to the number of samples selected per active learning iteration, is one solution to enhance computational efficiency (Citovsky et al., 2021; Zhang et al., 2022). However, it weakens the performance of active learning, and there remains the considerable computational cost associated with training the entire network multiple times. Therefore, recent research has explored single-shot active learning based on pre-trained features (Xie et al., 2023). However, as the number of labels increases, the effectiveness of this method gradually diminishes in comparison to other standard active learning strategies.

Another solution to boost active learning efficiency is to use less computationally intensive models or training methods as proxy tasks for sample selection, known as Selection via Proxy (SVP) (Coleman et al., 2019). Typical proxy tasks include reducing model complexity, such as using models like ResNet-8 or ResNet-14, early stopping, or training linear classifiers or MLPs based on pre-trained features (Zhang et al., 2024). However, these proxy tasks often have an impact on active learning performance and may result in savings in computation costs not outweighing the increase in annotation costs. As a result, practitioners are often faced with a trade-off between computational efficiency and the overall cost.

## 3 Preliminary

### 3.1 Selection via Proxy based on Pre-computed Features

We briefly review the Selection via Proxy framework based on pre-computed features, SVPp, as shown in fig. 3. We denote the labeled set as $L$, the unlabeled set as $U$. The pre-trained neural network backbone is defined as $f(\cdot; \theta_p) : \mathcal{X} \to \mathbb{R}^d$, where $\theta_p$ represents weights of the pre-trained model, $\mathcal{X}$ is the data space and $\mathbb{R}^d$ is feature space. And we define the predictor as $h(\cdot; \theta_h) : \mathbb{R}^d \to \mathbb{R}^c$, where $\theta_h$ is weights of the predictor

and $\mathbb{R}^c$ is the output space. SVPp follows a three-step process. First, all samples are forward-passed through $f(\cdot; \theta_p)$, and the pre-trained features are saved. Subsequently, the proxy model is trained, and samples are selected iteratively based on the active learning strategy. In this process, the predictor $h(\cdot; \theta_h)$ serves as the proxy model, utilizing pre-computed features as its input. The final step is fine-tuning the pre-trained model $h(f(\cdot; \theta_p); \theta_h)$ based on the obtained labels.

Figure 3: The Selection via Proxy based on pre-trained features (SVPp) Framework. Stage 1: pre-computing features. Stage 2: sample selection based on the proxy model (a simple classifier) with pre-computed features as input. Stage 3: Fine-tuning the pre-trained model using labeled samples.

## 3.2 LogME-PED

LogME-PED is an efficient metric for selecting pre-trained models, which outputs a score reflecting the performance of fine-tuned models. **LogME** uses evidence as a metric to assess the compatibility between a given pre-trained feature and the labels of the training samples (You et al., 2021), as shown in equation 1, where $F$ denotes the pre-trained features of the labeled set and $w$ denotes the parameters of the linear classifier. For specific calculation methods, refer to (You et al., 2021).

$$p(y|F) = \int p(w)p(y|w, F)dw \tag{1}$$

**PED** is a physically inspired model, that simulates feature learning dynamics through multiple iterations (Li et al., 2023). The approach draws an analogy between the optimization objective during fine-tuning and the concept of potential energy in physics. In fine-tuning, adjusting the model parameters forces the features of different classes to be distinguished, akin to the movement of objects influenced by forces in physics. Following this intuition, PED models the features of different classes as separate small balls, where the mean and standard deviation of each class's features correspond to the position and radius of the ball, respectively. The force between balls is proportional to their overlapping length. In this physical model, the positions and forces of the objects are updated iteratively until convergence. Finally, the updated features are regarded as the features after fine-tuning the pre-trained model. Then, LogME is computed based on these updated features to assess the performance of the fine-tuned model.

## 4 Observation and Analysis

Since practitioners utilizing proxy models for efficient active learning aim to reduce sample selection time while minimizing the loss of active learning gains (between standard active learning based on fine-tuned models and SVPp), the exact overlap between samples selected by the proxy model and those selected by the fine-tuned model is not their primary concern. Therefore, in this section, we first investigate **which differences in sample selection lead to the degradation of SVPp active learning performance**.

### 4.1 Impact of Sample Selection Differences on SVPp Performance

We categorize the samples selected by both the proxy model and the fine-tuned model into different regions based on whether the models can correctly predict the samples and the active learning metrics (such as margin (Scheffer et al., 2001), entropy (Lewis & Catlett, 1994), etc.). As shown in fig. 4, the x and y axes represent whether the proxy model and the fine-tuned model can correctly predict the samples, respectively. The closer a sample is to the origin, the more likely it is to be selected in active learning. For uncertainty-based active learning strategies (e.g., margin, entropy, confidence (Wang & Shang, 2014)) and strategies combining uncertainty and diversity (e.g., BADGE (Ash et al., 2020)), a smaller distance from the origin indicates higher uncertainty, such as a smaller margin predicted by the model. For distance-based active learning strategies (e.g., coreset (Sener & Savarese, 2018)), a smaller distance from the origin indicates a greater distance between the sample and the labeled set. Most effective active learning strategies for fine-tuning pre-trained models are uncertainty-based or combine uncertainty with diversity (where the latter extends the former by selecting diverse samples from the uncertainty candidates) (Zhang et al., 2024; Emam et al., 2021), this paper will use uncertainty-based active learning as an example for analysis.

Standard active learning based on the fine-tuned model selects samples from regions O, A, and B, while efficient active learning based on the proxy model selects samples from regions O, C, and D. In other words, compared to the fine-tuned model, the proxy model misses samples from regions A1, A2, B1, and B2 while additionally selecting samples from regions C and D.

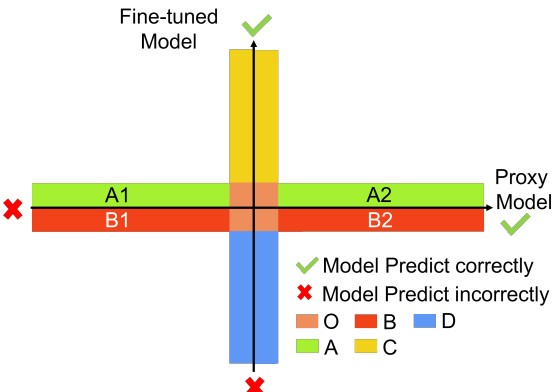

Figure 4: Sample Selection Discrepancy: Proxy vs. Fine-tuned Models. For instance, in uncertainty-based active learning strategies, the two axes represent the confidence of predictions for the proxy model and the fine-tuned model, with the positive half-axis indicating correct predictions and the negative half-axis indicating incorrect predictions. AL strategy selects samples with the lowest prediction confidence, meaning the proxy model chooses samples from regions O, C, and D, while the fine-tuned model selects samples from regions O, A, and B. Depending on whether the proxy model correctly predicts the samples, regions A and B are further divided into subregions A1, A2, B1, and B2. Both the proxy model and the fine-tuned model confidently predict the remaining white region, hence they do not select samples from it.

**Empirical Evidence**   We conducted replacement experiments to demonstrate the impact of missing samples from regions A1, A2, B1, and B2 on SVPp active learning performance. In this section, we used one of the state-of-the-art strategies, margin sampling, with fine-tuned pre-trained models. A total of 400k samples were selected over 16 rounds, with the first 10k samples selected randomly. The experiments were conducted on ImageNet, with detailed settings provided in sec. 6.1.

Specifically, during each round of sample selection, we replaced the tail samples selected by the proxy model (i.e., those with the largest margins) with samples from regions A1, A2, B1, and B2. As shown in fig. 5, we observed that not all differences in sample selection caused performance degradation in SVPp active learning. Replacing samples with those from regions A2 and B2 improved SVPp active learning performance, replacing samples from region A1 had no effect, while replacing samples from region B1 decreased performance. Based

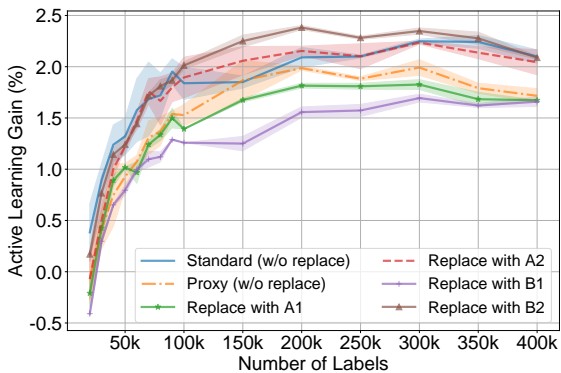

Figure 5: The impact of different regions on SVPp active learning performance. Replacing samples selected by the proxy model with those from regions A1, A2, B1, B2. The active learning gain refers to the difference in accuracy between active learning and the random baseline.

on these observations, to improve the performance of SVPp active learning, we should focus on the sample selection differences from regions A2 and B2.

To investigate why samples from regions A1 and B1 fail to improve SVPp performance, we analyzed the difficulty of samples across different regions. Recent studies suggest that hard-to-learn samples are likely to undermine active learning performance (Karamcheti et al., 2021). A1 and B1 regions, compared to A2 and B2, likely contain a higher proportion of difficult samples. This is because the fine-tuned model's predictions on these samples show high uncertainty, while the proxy model's predictions on them are wrong. To validate this hypothesis, we utilized Dataset Maps (Swayamdipta et al., 2020) to assess sample difficulty. Dataset Maps depict sample learnability by analyzing training dynamics from models trained on the entire dataset, using metrics such as average confidence for the correct class and prediction accuracy across training epochs to quantify difficulty. Following Karamcheti et al. (2021), we categorized sample difficulty based on mean confidence levels: very hard ($<0.25$), hard ($>=0.25$ and $<0.5$), medium ($>=0.5$ and $<0.75$), and easy ($>=0.75$). As illustrated in fig. 6, regions A1 and B1 indeed contain a higher proportion of very hard samples.

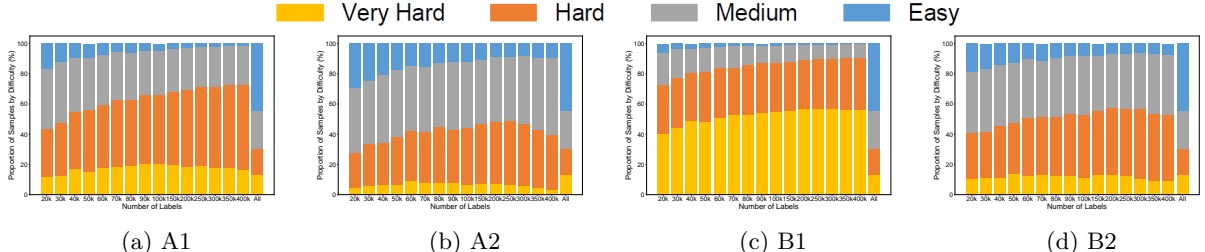

(a) A1        (b) A2        (c) B1        (d) B2

Figure 6: Difficulty distribution of samples from different regions.

## 4.2 Role of Region A2 and B2 in Improving SVPp Performance

Those missed samples (Region A2, B2) that can improve SVPp performance share a common characteristic: the proxy model can confidently predict them correctly, while the fine-tuned model shows high uncertainty in its predictions. This indicates that these samples are well-distinguished using pre-trained features, whereas they are harder to differentiate in the fine-tuned features. Including these samples in the next round of active learning will help the fine-tuned model distinguish them. A question arises: **are the pre-trained feature manifold and the fine-tuned feature manifold (training with samples from regions A2 and B2) similar?**

**Empirical Evidence** To answer this question, we conducted a backbone exchange experiment to compare the similarity between the fine-tuned and pre-trained features. We put the linear classifier of the fine-tuned model on top of the pre-trained backbone and evaluated its accuracy loss on the test set. The greater the accuracy drop after exchanging the backbone, the more dissimilar the fine-tuned features are from the pre-trained features.

We created five sets of training samples by replacing 0% (w/o replacing), 5%, 10%, 15%, and 20% of the samples selected by the proxy model with the samples from regions A2 and B2 that had the smallest margins. After training on these sets, the accuracy loss is evaluated by putting the fine-tuned model's linear classifier on top of the pre-trained backbone. As shown in fig. 7, the accuracy drop decreased as more samples from A2 and B2 were included in the train set, indicating that incorporating these samples helps maintain the feature manifold similar to the pre-trained model.

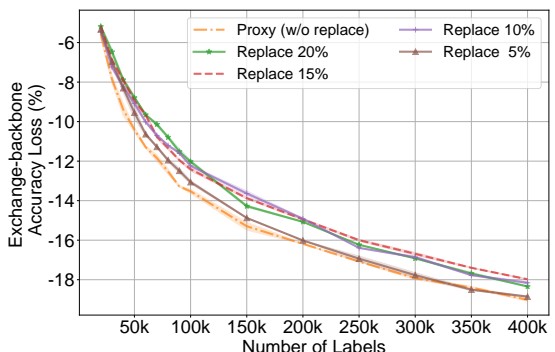

Figure 7: The influence of samples from regions A2 and B2 on the similarity between fine-tuned and pre-trained features. Exchange-backbone accuracy loss refers to the accuracy difference between the fine-tuned model's linear classifier placed on the pre-trained backbone and the fine-tuned model.

Figure 8: Comparison of active learning performance based on the fine-tuned model (standard active learning) and proxy model when employing different training methods. The active learning gain refers to the difference in accuracy between active learning and the random baseline (using fine-tuning).

Based on this observation, another intriguing question arises: **can directly maintaining the similarity between fine-tuned and pre-trained features mitigate the performance drop caused by the proxy model missing samples from regions A2 and B2?**

**Empirical Evidence** To explore this, we employed two training methods: fine-tuning and LP-FT (Linear Probing and then Fine-Tuning), to compare the performance differences between SVPp (sample selection via proxy model and evaluation via fine-tuning or LP-FT) and standard active learning (both sample selection and evaluation via fine-tuning or LP-FT). In LP-FT, linear probing is followed by fine-tuning to preserve the similarity between the fine-tuned features and the pre-trained features.

As shown in fig. 8, after applying LP-FT, the active learning performance based on the proxy model was almost identical to that based on the LP-FT model. This suggests that samples from regions A2 and B2 enhance active learning performance by preserving the similarity between the fine-tuned and pre-trained feature manifold. Therefore, directly preserving this similarity compensates for the decrease in active learning performance caused by the proxy model's omission of samples from regions A2 and B2.

## 4.3 Discussions

The previous section discussed that not all differences in sample selection between SVPp and standard active learning result in performance differences (only those from regions A2 and B2 do). Moreover, when using LP-FT to train the model, the sample selection differences from regions A2 and B2 do not cause a decline in SVPp active learning performance. Next, we discuss the limitations of the above analysis.

Firstly, given the training samples, we always aim to choose the training method that achieves the highest accuracy. Therefore, it is only meaningful to use LP-FT to compensate for the proxy model missing samples from regions A2 and B2 if LP-FT can achieve accuracy that is at least as good as fine-tuning. Secondly, as the number of selected samples increases, the performance gap between the fine-tuned model and the proxy model may widen (Cai et al., 2021). This could reduce the active learning gain of SVPp, thus affecting the previous conclusion about the impact of missed regions (A1, A2, B2, B2) by the proxy model on active learning performance. Therefore, we further analyzed the proportion of samples from regions C and D, as well as their impact on active learning gain.

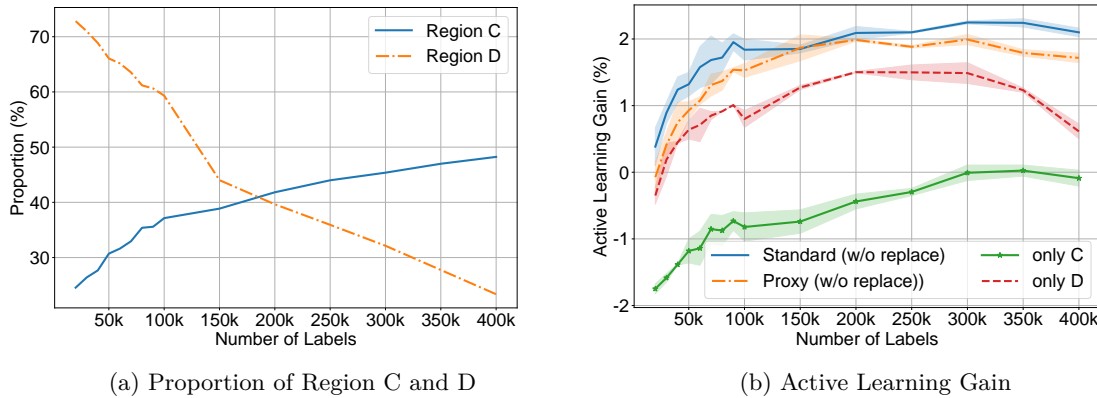

(a) Proportion of Region C and D        (b) Active Learning Gain

Figure 9: Observations on region C and D: (a) the proportion of region C and D, (b) comparison of active learning gain when selecting only samples from region C or D. The active learning gain refers to the difference in accuracy between active learning and the random baseline.

As shown in fig. 9a, with more samples selected during active learning, the proportion of samples from region C selected by the proxy model increases, while the proportion from region D decreases. This aligns with intuition: as the number of annotations increases, the features of the fine-tuned model become better at distinguishing between sample categories than those of the proxy model, increasing correctly predicted samples (samples from region C). Additionally, as shown in fig. 9b, selecting samples from region C results in an active learning gain of less than 0, meaning it performs worse than random selection. These two observations imply that as more samples are selected during active learning, the proxy model selects more samples from region C, causing its active learning performance to decline. As the proxy model's active learning performance declines, it may lead to all missed regions (A1, A2, B1, B2) contributing to performance differences between SVPp and standard active learning.

In summary, when LP-FT training achieves performance comparable to fine-tuning and the proportion of samples from region C is small, using LP-FT can prevent the decline in active learning gains caused by SVPp. The next section discusses how to adjust proxy model-based active learning to build an efficient active learning framework when these conditions are not met.

## 5 Method

This section first addresses the issue of the increasing proportion of region C samples, which makes LP-FT insufficient to prevent the decline in SVPp active learning performance. Region C refers to samples that the fine-tuned model can predict correctly with high confidence, while the proxy model struggles to differentiate them (high uncertainty). This indicates that the fine-tuned model's features are more suitable for distinguishing these samples. To address this, we recommend fine-tuning the pre-trained model as the number of region C samples increases. Updating the pre-computed features enhances the proxy model's discriminative capacity, thereby reducing the proportion of region C samples. Additionally, when updating pre-computed features, we switch the model training method from LP-FT to fine-tuning. Continuing with LP-FT when fine-tuned features outperform pre-trained features can lead to suboptimal results by hindering necessary modifications to the pre-trained model.

The ASVP framework is divided into three stages: pre-computation of features, data selection, and final model training. The data selection stage of our method is illustrated in fig. 10. During the data selection stage, the proxy model uses pre-computed features (pre-computed features refer to pre-trained features when feature updating is not triggered) as input and performs multiple rounds of active learning (alternating between sample selection and proxy model training). At each iteration, we check if an update to the pre-computed features is needed. If the annotation budget is exhausted without triggering a feature update, LP-FT is used to train the final model. If an update is detected, fine-tuning is performed to generate new pre-computed features for the following rounds of sample selection. After the sample selection, the final model is trained using fine-tuning.

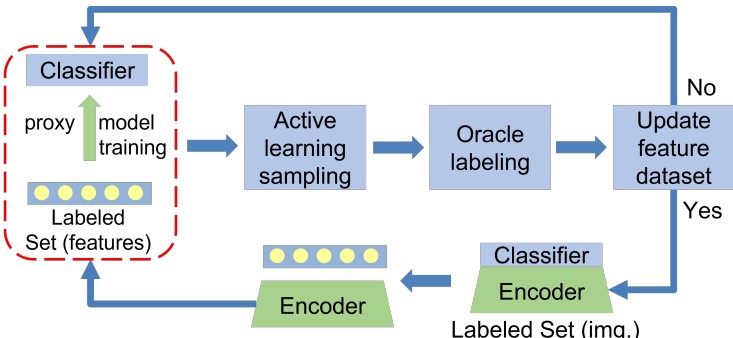

Figure 10: The Data Selection Pipeline in our approach, Aligned Selection via Proxy (ASVP). After acquiring new labels from the oracle, we assess the necessity of updating the pre-computed features. If required, the pre-trained model is fine-tuned, and the pre-computed features are updated.

Next, we explain when it becomes necessary to update the pre-computed features. As discussed in sec. 4, as the number of annotations increases, the quality of fine-tuned features improves gradually, resulting in a higher proportion of samples selected by the proxy model from the redundant region. Also, including the samples from the redundant region rarely improves the performance of the fine-tuned model. Hence, we assess the ratio of redundant region samples by evaluating the changes in the performance of the fine-tuned model after adding more annotated samples. To assess the changes in the fine-tuned model's performance, we employ LogME-PED (You et al., 2021; Li et al., 2023) as an indicator.

Specifically, in every active learning iteration, we first compute the PED component. If convergence is achieved within a single iteration, we infer that the fine-tuned features will maintain similarity to the pre-trained features. So, there is no need to calculate the LogME metric and update the pre-computed features. However, if the PED component converges with more than one iteration, we proceed to compute the LogME metric. When the difference between the current LogME score and that from the previous active learning iteration is below a predefined threshold, we deduce that the fixed pre-trained features pose a significant obstacle to the performance improvement of the proxy model, prompting us to fine-tune the model and update the pre-computed features.

# 6 Results

Our method is validated on four datasets: ImageNet-1k (Deng et al., 2009), CIFAR-10 (Krizhevsky et al., 2009), CIFAR-100 (Krizhevsky et al., 2009) and Oxford-IIIT Pet dataset (Parkhi et al., 2012) with five typical active learning strategies. The experiment setup is clarified in sec. 6.1. We propose the new efficient active learning evaluation metric in sec. 6.2. The results are shown in sec. 6.3 and sec. 6.4. The ablation study is shown in sec. 6.5. The detailed study of the impact of the position of updating features is shown in sec. 6.6. Additionally, given the efficiency of our framework, decreasing the amount of sample selected in each active learning iteration to alleviate the redundant problem of batch active learning strategy is feasible. The results are shown in appendix F.

### 6.1  Experiment Setup

**Active learning strategies.** To evaluate the compatibility of our proposed ASVP with existing active learning strategies, four typical active learning strategies are included: (1) Margin: uncertainty-based sampling, one of the SOTA in the context of fine-tuning pre-trained models (Scheffer et al., 2001), (2) Confidence: uncertainty-based sampling (Wang & Shang, 2014), (3) Coreset: diversity-based sampling (Sener & Savarese, 2018), (4) BADGE: combination of uncertainty and diversity (Ash et al., 2020), (5) ActiveFT(al), feature space coverage with features from fine-tuned model (Xie et al., 2023).

**Baseline.** (1) Standard active learning: training the whole model with fine-tuning (FT) or linear-probing then fine-tuning (LP-FT) (Kumar et al., 2022) in each active iteration, (2) SVPp (Zhang et al., 2024): selecting samples via MLP proxy model based on pre-trained features and training the final model with FT, (3) ActiveFT(pre) (Xie et al., 2023): a single-shot active learning strategy that selects all sample in single iteration based on pre-trained features.

**Implementations.** We utilized the ResNet-50 model (He et al., 2016), pre-trained on ImageNet using the BYOL-EMAN (Cai et al., 2021) method in all our experiments. The training hyper-parameters of the two model training approaches, fine-tune (FT) and LP-FT, are provided in appendix A. For the baseline method (SVPp) and our method (ASVP), we employed 2-layer MLPs as proxy models, where its architecture is Linear + BatchNorm + ReLU + Linear. The hidden layer width matched the input feature dimensions. We performed multiple active learning iterations until the model performance approached that of training on the entire dataset (upper bound). Specifically, for the ImageNet, CIFAR-10, CIFAR-100, and Pets datasets, we conducted 16, 18, 22, and 10 active learning iterations, respectively. The query step was set to 200 samples per iteration for the Pets dataset. For ImageNet, CIFAR-10, and CIFAR-100, we used varying query steps: for ImageNet, we queried 10,000 samples per iteration until reaching 100,000 samples, after which we queried 50,000 samples per iteration. For CIFAR-10, we queried 200 samples per iteration until reaching 2,000 samples, then increased to 500 samples per iteration. For CIFAR-100, we queried 400 samples per iteration until reaching 6,000 labeled samples, then switched to 2,000 samples per iteration. The difference in LogME-PED scores between consecutive active learning iterations is set as 1 for the threshold of updating pre-computed features.

### 6.2  Metric

Efficient active learning methods often exhibit a lower accuracy compared to standard active learning approaches. Understanding the additional annotation costs needed to compensate for the accuracy gap caused by efficient active learning allows practitioners to weigh the computational cost savings against increased labeling expenses, aiding their decision to utilize these methods.

To this end, we propose a new evaluation metric: equivalent saving amount. Let $N_1$ represent the number of samples selected by the active learning strategy, and let $A$ be the accuracy achieved by a model trained with these $N_1$ labeled samples. We utilize interpolation to estimate the number of samples, $N_2$, required to achieve accuracy $A$ for a random baseline. We refer to $N_2$ as the **equivalent number of non-AL labels** in the subsequent discussions. The number of saved samples is $N_2 - N_1$ and **sample saving ratio** is $\frac{N_2 - N_1}{N_2}$. Subsequently, we compute the **average sample saving ratio** across all iterations of active learning to assess the performance of the active learning strategy. Furthermore, the **overall cost** of active learning, including annotation and training costs, equals $N_1 \times P_l + C_{tr}$, where $P_l$ denotes the unit price of annotation and $C_{tr}$ represents the training cost of active learning.

### 6.3  Sample Saving Ratio and Overall Cost

The average sample savings ratios are presented in table 1, with each experimental setting repeated three times to report both the average and standard deviation. The table also illustrates the total cost required for active learning to reach the accuracy achieved by the random baseline at the last active learning iteration (i.e., the accuracy achieved when randomly selecting 400k, 6000, 20000, and 2000 samples from ImageNet, CIFAR-10, CIFAR-100, and Pets datasets, respectively). The training cost is assessed based on AWS EC2 P3 instances (Zhang et al., 2024), while the annotation cost is evaluated using AWS Mechanical Turk,

with three annotations per sample to ensure labeling quality. Additionally, Figures 11 and 12 show the accuracy and corresponding label savings achieved with margin sampling, while results using other active learning strategies are provided in Appendix E. Figure 13 shows the sampling time and total cost required by standard active learning, SVPp, ASVP (ours), and the single-shot active learning strategy, ActiveFT (pre), to achieve the accuracy attained by the random baseline at the final active learning iteration.

Table 1: Average sample savings ratios, with each configuration repeated three times to calculate the average and standard deviation. The training method indicates how the final model is trained, and the selection method indicates the model used for sample selection. The cost refers to the total cost required for active learning to achieve the accuracy achieved by the random baseline in the last active learning iteration. The cost-saving ratio refers to the average ratio of the cost of active learning to the cost of the random baseline. The best results are shown in red and the second-best results are in blue.

| Selection Method | Training Method | AL Strategy | ImageNet Avg. Sample Saving Ratio | Cost ($) | CIFAR-10 Avg. Sample Saving Ratio | Cost ($) | CIFAR-100 Avg. Sample Saving Ratio | Cost ($) | Pets Avg. Sample Saving Ratio | Cost ($) | Cost Saving Ratio |
|---|---|---|---|---|---|---|---|---|---|---|---|
| Standard | FT | Random | 0% | 14400 | 0% | 216 | 0% | 720 | 0% | 72 | 100% |
| Standard | FT | Margin | 30.30±1.09 | 8752 | 40.23±0.51 | 134 | 14.54±2.46 | 578 | 31.67±2.30 | 49 | 68% |
| | | BADGE | 21.33±0.38 | 10018 | 36.18±0.34 | 128 | 13.73±2.07 | 580 | 33.01±3.17 | 47 | 69% |
| | | Confidence | -3.42±0.75 | 9619 | 24.99±1.37 | 142 | 4.23±2.34 | 567 | 27.24±3.28 | 50 | 70% |
| | | Coreset | - | - | 12.44±2.42 | 192 | -13.90±2.31 | 722 | -4.05±2.31 | 74 | 97% |
| | | ActiveFT(al) | - | - | 12.39±3.26 | 186 | 3.03±1.17 | 798 | -5.20±2.28 | 97 | 111% |
| Standard | LP-FT | Margin | 32.22±0.27 | 9884 | 37.39±1.28 | 139 | 22.31±1.83 | 509 | 6.97±2.55 | 73 | 76% |
| | | BADGE | 24.12±0.54 | 11901 | 29.54±0.70 | 136 | 19.81±1.28 | 528 | 6.74±3.84 | 68 | 78% |
| | | Confidence | 0.11±1.51 | 10125 | 20.10±4.01 | 138 | 13.49±2.90 | 510 | 9.58±3.41 | 73 | 77% |
| | | Coreset | - | - | -4.60±4.09 | 244 | -16.76±1.78 | 769 | -36.93±4.53 | 110 | 124% |
| | | ActiveFT(al) | - | - | -2.13±0.11 | 249 | 6.38±1.35 | 755 | -29.33±10.58 | 104 | 122% |
| Single-shot | FT | ActiveFT(pre) | - | - | 1.40±1.50 | 235 | -0.21±1.41 | 822 | -11.92±1.26 | 118 | 129% |
| | LP-FT | | - | - | 5.02±4.02 | 238 | 8.92±2.34 | 742 | -38.10±2.61 | 235 | 180% |
| SVPp | FT | Margin | 25.16±1.47 | 9094 | 19.86±2.53 | 184 | 3.51±0.77 | 654 | 23.48±4.09 | 50 | 77% |
| | | BADGE | 16.17±1.84 | 11374 | 19.35±3.66 | 156 | 0.51±0.74 | 657 | -7.78±2.42 | 85 | 90% |
| | | Confidence | -8.56±2.88 | 9678 | 1.53±1.82 | 171 | -7.93±1.85 | 644 | 2.40±7.25 | 66 | 82% |
| | | Coreset | - | - | -54.78±8.88 | 1355 | -56.53±3.12 | 903 | -38.63±11.64 | 87 | 291% |
| | | ActiveFT(al) | - | - | 3.79±4.31 | 192 | -0.50±1.42 | 713 | -11.63±6.85 | 319 | 210% |
| ASVP | FT/LP-FT | Margin | 34.38±1.57 | 8690 | 32.45±1.13 | 127 | 12.82±1.40 | 590 | 21.98±0.50 | 47 | 67% |
| | | BADGE | 25.57±2.54 | 10305 | 32.87±0.84 | 137 | 10.11±1.13 | 577 | 6.10±5.59 | 55 | 73% |
| | | Confidence | 5.88±0.66 | 9479 | 25.67±2.48 | 167 | 2.75±1.17 | 616 | 14.81±3.17 | 50 | 75% |
| | | Coreset | - | - | 11.11±1.74 | 192 | -26.16±1.76 | 767 | -46.3±10.26 | 91 | 107% |
| | | ActiveFT(al) | - | - | 26.67±0.66 | 155 | 6.81±1.70 | 775 | -7.74±4.81 | 47 | 82% |

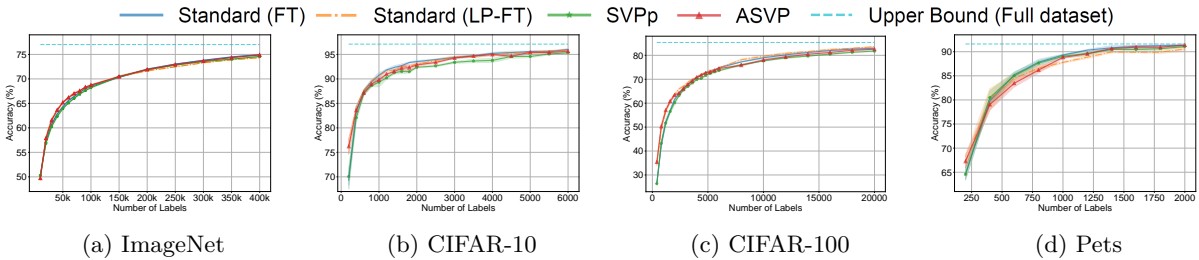

(a) ImageNet    (b) CIFAR-10    (c) CIFAR-100    (d) Pets

Figure 11: The accuracy of the standard method, SVPp, and our method ASVP on (a) ImageNet, (b) CIFAR-10, (C) CIFAR-100 and (d) Pets using margin sampling.

Our method consistently outperforms the efficient active learning method, SVPp, in terms of sample saving ratio and overall cost reduction. It surpasses the single-shot active learning strategy, ActiveFT(pre), when combined with most active learning strategies. In most cases, our approach offers similar or greater overall cost savings compared to standard active learning. This alleviates practitioners' concerns about overall costs when using efficient active learning. Additionally, in standard active learning methods, the lack of a clear criterion for choosing the proper training method may lead to suboptimal results. Specifically, the standard active learning with FT outperforms the standard active learning with LP-FT on CIFAR-10 and Pets, while on ImageNet and CIFAR-100, the standard (LP-FT) method demonstrates better performance.

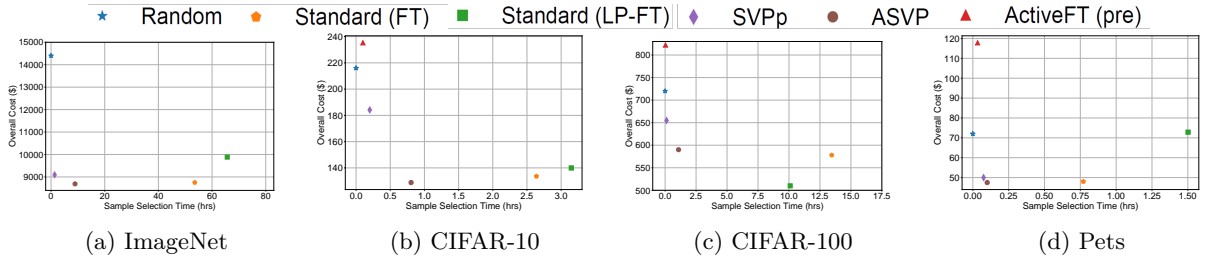

(a) ImageNet
(b) CIFAR-10
(c) CIFAR-100
(d) Pets

Figure 12: The equivalent number of non-active learning label amounts comparison of the standard method, SVPp, and our method ASVP on (a) ImageNet, (b) CIFAR-10, (C) CIFAR-100 and (d) Pets using margin sampling.

(a) ImageNet
(b) CIFAR-10
(c) CIFAR-100
(d) Pets

Figure 13: Computation efficiency and overall cost comparison of the standard active learning method, SVPp, and our method ASVP on (a) ImageNet, (b) CIFAR-10, (C) CIFAR-100 and (d) Pets using margin sampling.

In contrast, our approach, by switching training methods, ensures a relatively favorable performance in practical applications.

Compared to the single-shot active learning method ActiveFT(pre), our approach is compatible with various existing active learning strategies and exhibits lower dependence on the quality of pre-trained features. The reliance of ActiveFT(pre) on the quality of pre-trained features leads to suboptimal results. This is particularly evident in datasets like Pets, where the pre-trained feature quality is low.

## 6.4 Computational Time

We define the average training time for the fine-tuned and the proxy models in each active learning iteration as $T_{tr,full}$ and $T_{tr,proxy}$, respectively. The time for processing all unlabeled samples through the whole pre-trained model and the proxy model to compute active learning metrics and select samples is denoted as $T_{f,full}$ and $T_{f,proxy}$, respectively. The time spent on forwarding all samples through the model to record features is represented by $T_{pre}$. The number of active learning iteration is given by $N_{al}$. The total active learning time for standard active learning, $T_{s,full}$, is $N_{al} \times (T_{tr,full} + T_{f,full})$. The total time for SVPp, $T_{s,svpp}$, is $N_{al} \times (T_{tr,proxy} + T_{f,proxy}) + T_{pre}$. For ASVP, the initial pre-computed features are obtained from a pre-trained model. When updating these pre-computed features is necessary, the model is fine-tuned on labeled data, and the pre-computed features are refreshed accordingly. Let $N_{pre}$ denote the number of times the ASVP features are pre-computed, then, the time required for pre-computing features is $N_{pre} \times T_{pre} + (N_{pre}-1) \times T_{tr,full}$. The total time required by ASVP, denoted as $T_{s,asvp}$, is $N_{al} \times (T_{tr,proxy} + T_{f,proxy}) + N_{pre} \times T_{pre} + (N_{pre}-1) \times T_{tr,full}$.

The computation time of standard active learning, SVPp, and ASVP on ImageNet is presented in table 2. The results on other datasets are shown in appendix B. The time for ASVP mainly arises from updating pre-computed features, as the computation time for the proxy model, an MLP, is significantly lower than that of the fine-tuned model. This means that increasing the number of active learning iterations $N_{al}$ has a marginal impact on ASVP's total time. Therefore, this opens up the potential to improve the diversity of selected samples by reducing the number of samples chosen in each active learning iteration.

Table 2: Computation time comparison of various active learning sample selection methods on ImageNet using V100 GPU.

| | $T_{pre}(hrs)$ | $N_{al} \times T_f(hrs)$ | $N_{al} \times T_{tr}(hrs)$ | $T_s(hrs)$ |
|---|---|---|---|---|
| Standard | - | 16.36 | 80.97 | 97.33 |
| SVPp | 0.92 | 0.10 | 0.46 | 1.48 |
| ASVP | 8.60 | 0.10 | 0.46 | 9.16 |

## 6.5 Ablation Study

Our ablation study was conducted on CIFAR-10, CIFAR-100, and Pets, using 10, 15, and 10 active learning iterations, respectively, with each iteration querying 200, 400, and 200 samples. All other settings follow those described in sec. 6.1. The results with margin sampling are shown in Table 3, while additional ablation studies with other active learning strategies are provided in Appendix C. Across all three datasets, updating pre-computed features consistently led to a noticeable performance boost. Meanwhile, switching training methods improved performance on all datasets except for the Pets dataset, suggesting that our method may not precisely determine the optimal point for switching training methods. However, considering the practical scenario where we cannot pre-determine whether using FT or LP-FT for training is better, our approach still aids in selecting a preferable training method.

Table 3: Ablation Study. Comparison of the average sample saving ratio of only updating pre-computed features (training the final model using fine-tuning), solely switching training methods (using pre-trained features), and the complete ASVP approach (both updating pre-computed features and switching training methods) under different datasets with margin sampling.

| Update Pre-computed Features | Switch Training Methods | CIFAR-10 | CIFAR-100 | Pets |
|---|---|---|---|---|
| No | No | 23.63±4.01 | 1.36±1.45 | 23.48±4.09 |
| No | Yes | 24.96±2.74 | 7.69±1.56 | 18.45±3.89 |
| Yes | No | 31.42±2.65 | 6.98±2.03 | 27.01±0.51 |
| Yes | Yes | 35.03±1.89 | 13.31±1.86 | 21.98±0.50 |

## 6.6 Influence from Position of Updating Features

We examined the impact of the position of updating pre-computed features on CIFAR-10. The average savings ratio is presented in table 4. The choice of updating pre-computed features does indeed affect the final performance of ASVP. Nevertheless, across the six different positions explored in the experiments, ASVP consistently outperforms the method relying solely on pre-training features (SVPp). Furthermore, our proposed method for determining the update location based on LogME-PED proves to be effective. On CIFAR-10, this method selects the update of pre-computed features at 600 labels, ranking as the second-best among the six experimental positions, as shown in table 4. The positions of updating pre-computed features estimated through LogME-PED are outlined in appendix D.

Table 4: The influence of the position of updating pre-computed feature on average sample savings ratios on CIFAR-10. The best results are shown in red and the second-best results are in blue.

| Change at | Avg. Sample Saving Ratio | Change at | Avg. Sample Saving Ratio |
|---|---|---|---|
| w/o | 23.63±4.01 | | |
| 200 | 33.13±2.32 | 800 | 36.03±1.48 |
| 400 | 32.13±4.60 | 1000 | 32.75±1.91 |
| 600 | 35.03±1.89 | 1200 | 31.33±1.87 |

# 7 Conclusion

Active learning on pre-trained models offers label efficiency but requires significant computational time, whereas current efficient active learning methods tend to compromise a notable fraction of active learning performance, resulting in increased overall costs. To address this challenge, this paper attributes the reduced performance of SVPp to (1) the absence of samples necessary to preserve fine-tuned features similar to the pre-trained features and (2) an increased selection of redundant samples due to fine-tuned features becoming better than pre-trained features as the annotation increases. Building upon this analysis, we propose a simple and effective method, ASVP, to improve the effectiveness (in terms of accuracy) of efficient active learning with comparable or marginally increased computational time. Additionally, we introduce the sample savings ratio as a metric to assess the effectiveness of efficient active learning, providing a straightforward measure for labeling cost savings. Experiments show that our proposed ASVP saves similar or greater total costs in most cases while maintaining computational efficiency.

# 8 Limitations and Future Directions

Due to limited computational resources, this paper evaluates efficient active learning methods' labeling and training costs using a ResNet-50 model with a specific pre-training method, BYOL-EMAN (Grill et al., 2020; Cai et al., 2021). Although ResNet-50 serves as a practical choice for our experiments and provides a reliable baseline, larger models achieve higher accuracy and are often more helpful in practical applications (Zhai et al., 2022). Existing research shows that larger models exhibit better label efficiency, requiring fewer labeled samples to reach the same accuracy as smaller models (Chen et al., 2020b). This implies that in active learning, using larger models may reduce labeling costs while increasing training costs. Therefore, the total cost savings achieved by both standard and efficient active learning methods are closely tied to model size. Evaluating the impact of model size on total cost savings, and exploring efficient active learning methods that can effectively reduce sample selection time and total cost (labeling and training costs) in the context of larger models, are important directions for future research.

Furthermore, when transferring to downstream datasets (used for active learning), different pre-trained models exhibit varying performance gaps between fine-tuning and linear probing. A smaller gap suggests higher-quality pre-trained features, making it easier for proxy models based on these features to select samples similar to those chosen by fine-tuned models. This means that proxy model-based active learning may compromise less on performance compared to standard active learning. Therefore, exploring how to select suitable pre-trained models is another interesting direction for future research.

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

## A   Hyperparameters

Fine-tuning, linear probing then fine-tuning (LP-FT), and proxy model training all utilized the SGD with momentum optimizer, with a momentum value of 0.9. For fine-tuning and LP-FT, the batch size in the training stage is 128 and the batch size in the sample selection stage of the active learning is 512.

The hyper-parameters of fine-tuning are shown in table 5. For ImageNet, we adopted the fine-tuning parameters following the prior work (Cai et al., 2021). Setting the learning rates separately for the classifier head and the backbone. For CIFAR-10, CIFAR-100, and Pets, we followed prior work (Cai et al., 2021) to set the learning rate of the classifier as 0.1 and to search other hyper-parameters. The hyper-parameters are detailed in table 5. The learning rate of the backbone was searched from (0.01, 0.05, 0.1 0.3) and the weight decay from (0, 0.0001). Additionally, to maintain consistency in training iterations between fine-tuning and LP-FT, we set the number of fine-tuning epochs as the sum of linear probing and fine-tuning epochs in LP-FT training.

Table 5: The hyper-parameters of the fine-tuning.

| Dataset | Learning Rate | | Training | Weight |
|---------|------------|----------|----------|--------|
|         | Classifier | Backbone | Epochs   | Decay  |
| ImageNet  | 0.1 | 0.01 | 50  | 0.0001 |
| CIFAR-10  | 0.1 | 0.1  | 130 | 0      |
| CIFAR-100 | 0.1 | 0.05 | 150 | 0      |
| Pets      | 0.1 | 0.1  | 140 | 0      |

For the LP-FT hyper-parameters, we conducted a search within the following ranges: (0.01, 0.1) for the classifier's learning rate, (0.01, 0.05, 0.1, 0.3) for the backbone's learning rate, (0.05,0.1,0.5) for learning rate in linear-probing (LP) stage, and (0, 0.0001) for weight decay. Fine-tuning (FT) training epochs were set at 120. For ImageNet, linear probing (LP) training epochs were searched from (5, 10), while for other datasets, LP training epochs were explored from (5, 10, 20, 30). Given the numerous hyper-parameters in LP-FT, we

sequentially search the backbone learning rate, classifier learning rate, weight decay, and LP training epochs. The final hyper-parameters are shown in table 6, where the weight decay is 0 and the learning rate in the LP stage is 0.5 across all datasets.

Table 6: The hyper-parameters of LP-FT.

| Dataset | Learning Rate | | Training Epochs | |
|---|---|---|---|---|
| | Classifier | Backbone | LP | FT |
| ImageNet | 0.01 | 0.01 | 5 | 45 |
| CIFAR-10 | 0.1 | 0.01 | 10 | 120 |
| CIFAR-100 | 0.1 | 0.01 | 30 | 120 |
| Pets | 0.1 | 0.1 | 20 | 120 |

The proxy model was trained 50 epochs with a learning rate of 0.01 and the weight decay of 0. During the training stage, the batch size is 2048 and in the active learning stage, the batch size is 16384.

## B  Computation Time

The time for standard active learning, SVPp and ASVP on V100 GPU is presented in table 7. The active learning setup, outlined in sec. 6.1, performs a total of 10 active learning iterations on CIFAR-10 and Pets, where 200 samples are selected for each iteration. Regarding CIFAR-100, it spans 15 iterations, selecting 400 samples per iteration.

Table 7: Sample selection time of standard active learning methods, SVPp, and our proposed method ASVP on CIFAR-10, CIFAR-100, and Pets. Experiments conducted on V100 GPU.

| | CIFAR-10 (hrs) | CIFAR-100 (hrs) | Pets (hrs) |
|---|---|---|---|
| Standard | 6.94 | 20.96 | 1.50 |
| SVPp | 0.22 | 0.13 | 0.03 |
| ASVP | 0.85 | 1.10 | 0.05 |

## C  Ablation Study

The results of ablation experiments with various active learning strategies are shown in table 8, table 9, table 10. Specifically, feature alignment refers to whether the ASVP method updates pre-computed features using fine-tuned model features, while training alignment indicates the selection of the final model training method (FT or LP-FT) based on LogME-PED scores. In all ablation experiments, feature alignment consistently demonstrates notable improvements. Training alignment, except for the Pets dataset, exhibits enhancements.

Table 8: Ablation Study: Comparing the average sample saving ratio using the active learning strategy **BADGE**.

| Feature alignment | Training alignment | CIFAR-10 | CIFAR-100 | Pets |
|---|---|---|---|---|
| No | No | 14.96±5.89 | -2.11±0.85 | -7.78±2.42 |
| No | Yes | 18.3±4.68 | 3.81±0.44 | -15.61±1.93 |
| Yes | No | 30.34±2.06 | 2.25±2.25 | 13.93±5.21 |
| Yes | Yes | 33.67±0.84 | 8.16±1.98 | 6.1±5.59 |

## D  Postion of Updating Pre-computed Features

The ASVP method uses the difference in LogME-PED scores between consecutive active learning iterations to decide if updating pre-computed features is required. The absolute differences in LogME-PED scores

Table 9: Ablation Study: Comparing the average sample saving ratio using the active learning strategy **confidence**.

| Feature alignment | Training alignment | CIFAR-10 | CIFAR-100 | Pets |
|:---:|:---:|:---:|:---:|:---:|
| No | No | -11.54±2.3 | -14.43±2.49 | 2.4±7.25 |
| No | Yes | -9.61±1.43 | -7.86±2.16 | -0.83±7.34 |
| Yes | No | 21.99±3.32 | -7.19±1.69 | 18.05±2.92 |
| Yes | Yes | 23.92±5.16 | -0.62±1.43 | 14.81±3.17 |

Table 10: Ablation Study: Comparing the average sample saving ratio using the active learning strategy **ActiveFT(al)**.

| Feature alignment | Training alignment | CIFAR-10 | CIFAR-100 | Pets |
|:---:|:---:|:---:|:---:|:---:|
| No | No | -4.72±5.78 | 0.25±0.92 | -11.63±6.85 |
| No | Yes | -1.89±6.12 | 5.66±0.97 | -14.76±6.74 |
| Yes | No | 24.62±1.31 | 5.86±2.24 | -4.61±4.97 |
| Yes | Yes | 27.45±1.42 | 11.28±2.01 | -7.74±4.81 |

are illustrated in fig. 14a, fig. 14b, fig. 14c. When the threshold is defined as 1, the updating positions are detailed in table 11.

Table 11: Positions of updating pre-computed features. The number of the PED convergence iterations is used to determine if updating pre-computed features is required. The positions are estimated by the absolute difference in LogME-PED scores between consecutive active learning iterations, where a difference smaller than 1 indicates an update.

| | Updating Features | Position |
|:---:|:---:|:---:|
| ImageNet | Yes | 150k |
| CIFAR-10 | Yes | 600 and 2000 |
| CIFAR-100 | Yes | 2000 and 6000 |
| Pets | Yes | 800 |

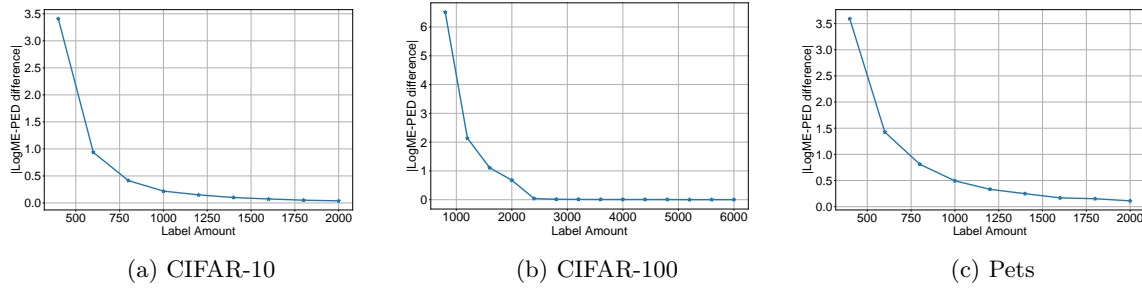

(a) CIFAR-10            (b) CIFAR-100            (c) Pets

Figure 14: The absolute differences in LogME-PED scores between consecutive active learning iterations on (a) CIFAR-10, (b) CIFAR-100 and (c) Pets.

# E    Results of other Active Learning Strategies

The accuracy and the equivalent non-active learning label amount of all active learning strategies on ImageNet, CIFAR-10, CIFAR-100 and Pets are shown in the fig. 15-fig. 22. The equivalent non-active learning label amount refers to the number of samples selected by a random baseline to achieve the same accuracy as active learning.

In fig. 23-fig. 26, we illustrate the overall costs (the labeling cost and the active learning training cost) and computational efficiency achieved by the standard active learning method, SVPp, and our proposed ASVP across different datasets. Compared to the existing efficient active learning method, SVPp, our method significantly improves the overall cost while maintaining the computational efficiency. ASVP achieves comparable overall costs while significantly reducing active learning computation time compared to the standard active learning method. This effectively addresses the trade-off practitioners encounter between computational efficiency and total costs, offering confidence in adopting efficient active learning methods.

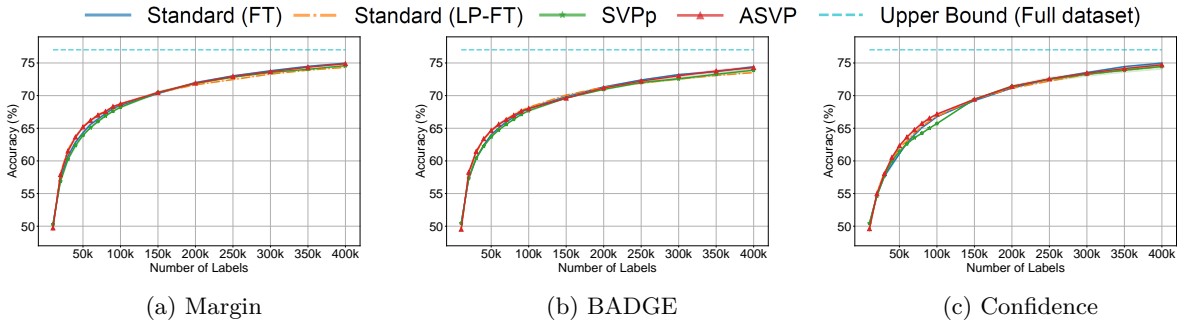

(a) Margin           (b) BADGE           (c) Confidence

Figure 15: The accuracy of the standard method, SVPp, and our method ASVP on the **ImageNet** using (a) Margin, (b) BADGE and (c) Confidence sampling.

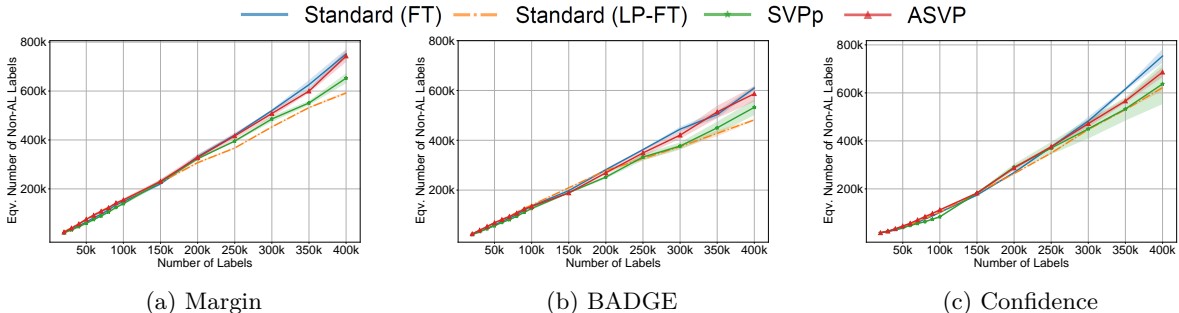

(a) Margin           (b) BADGE           (c) Confidence

Figure 16: The equivalent number of non-active learning label amounts comparison of the standard method, SVPp, and our method ASVP on **ImageNet** using (a) Margin, (b) BADGE and (c) Confidence sampling. The equivalent non-active learning label amount refers to the number of samples selected by a random baseline to achieve the same accuracy as active learning.

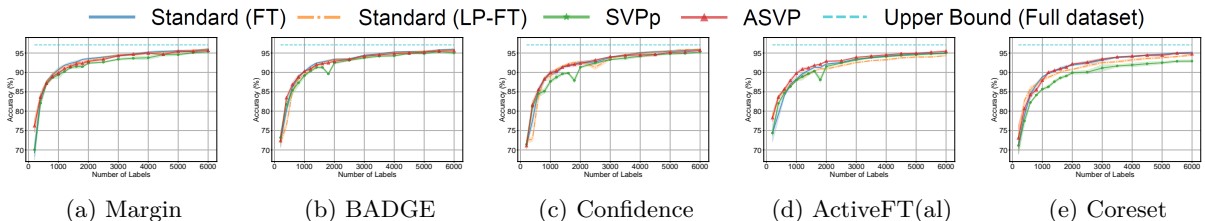

(a) Margin     (b) BADGE     (c) Confidence     (d) ActiveFT(al)     (e) Coreset

Figure 17: The accuracy of the standard method, SVPp, and our method ASVP on the **CIFAR-10** using (a) Margin, (b) BADGE, (c) Confidence, (d) ActiveFT(al) and (e) Coreset sampling.

## F   Improvement from Small Batchsize

The training costs associated with deep neural networks often lead practitioners to increase the number of samples selected per active learning iteration to ensure computational feasibility. Unfortunately, this

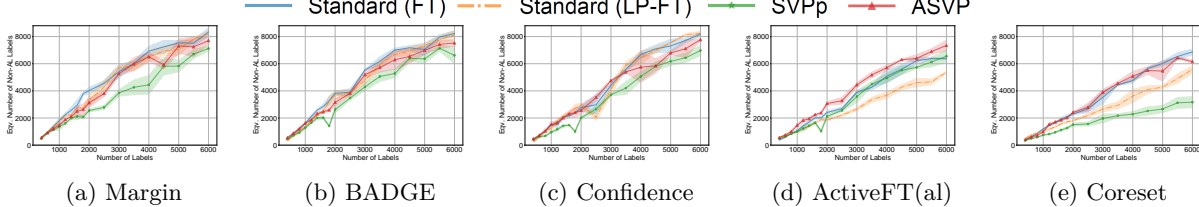

Figure 18: The equivalent number of non-active learning label amounts comparison of the standard method, SVPp, and our method ASVP on the **CIFAR-10** using (a) Margin, (b) BADGE, (c) Confidence, (d) ActiveFT(al) and (e) Coreset sampling. The equivalent non-active learning label amount refers to the number of samples selected by a random baseline to achieve the same accuracy as active learning.

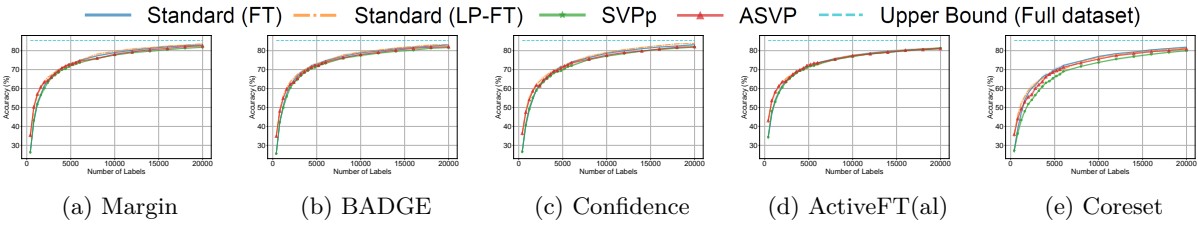

Figure 19: The accuracy of the standard method, SVPp, and our method ASVP on the **CIFAR-100** using (a) Margin, (b) BADGE, (c) Confidence, (d) ActiveFT(al) and (e) Coreset sampling.

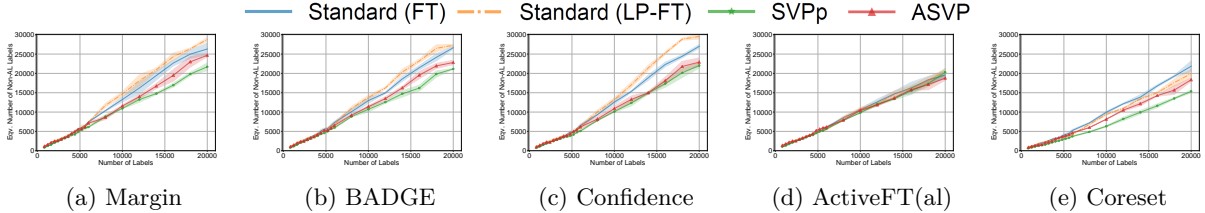

Figure 20: The equivalent number of non-active learning label amounts comparison of the standard method, SVPp, and our method ASVP on the **CIFAR-100** using (a) Margin, (b) BADGE, (c) Confidence, (d) ActiveFT(al) and (e) Coreset sampling. The equivalent non-active learning label amount refers to the number of samples selected by a random baseline to achieve the same accuracy as active learning.

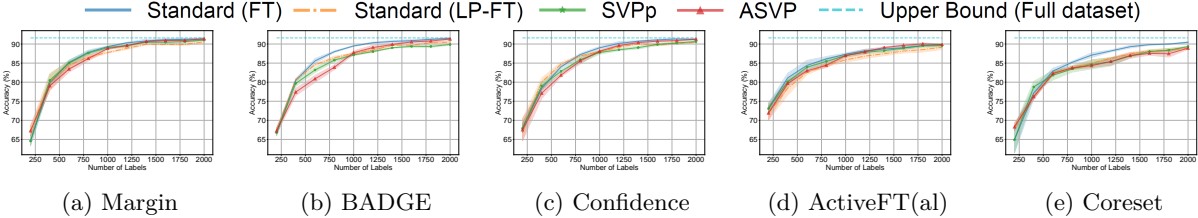

Figure 21: The accuracy of the standard method, SVPp, and our method ASVP on the **Pets** using (a) Margin, (b) BADGE, (c) Confidence, (d) ActiveFT(al) and (e) Coreset sampling.

strategy tends to result in selecting some redundant samples. Given the efficiency of our method, we can alleviate this problem. We conducted experiments on ImageNet, randomly selecting an initial pool of 10,000 samples. Subsequently, we performed 9, 45, 180, and 900 active learning iterations using margin sampling, selecting a total of 100,000 samples. The accuracy of the final model and the total sample selection time for both FT and LP-FT training are presented in table 12. The results show a general improvement in active learning performance with an increase in the number of iterations. However, beyond 180 iterations, the

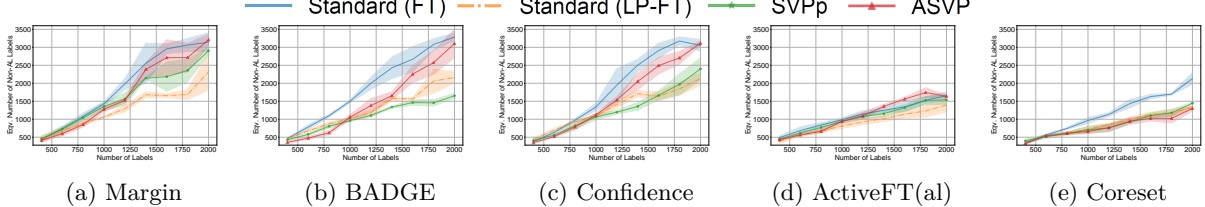

Figure 22: The equivalent number of non-active learning label amounts comparison of the standard method, SVPp, and our method ASVP on the **Pets** using (a) Margin, (b) BADGE, (c) Confidence, (d) ActiveFT(al) and (e) Coreset sampling. The equivalent non-active learning label amount refers to the number of samples selected by a random baseline to achieve the same accuracy as active learning.

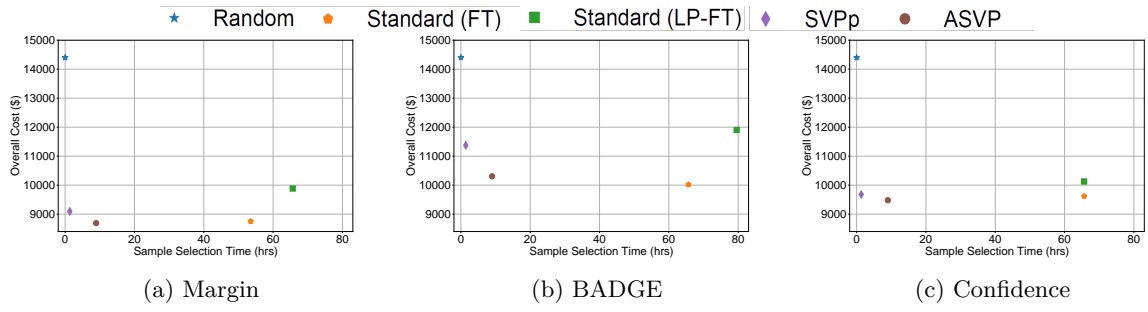

Figure 23: Computation efficiency and overall cost comparison of the standard active learning method, SVPp, and our method ASVP on **ImageNet** using (a) Margin, (b) BADGE and (c) Confidence sampling.

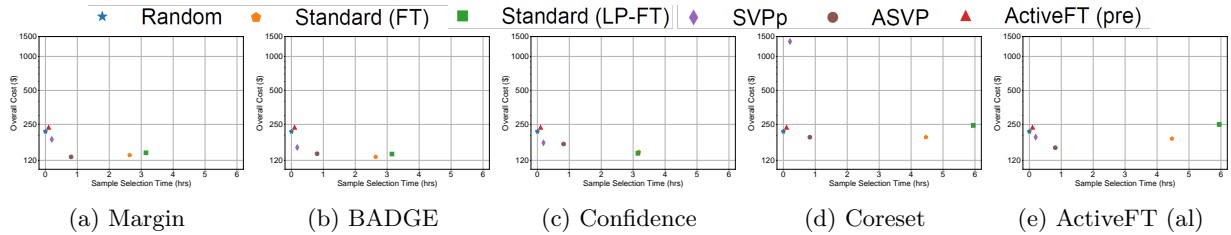

Figure 24: Computation efficiency and overall cost comparison of the standard active learning method, SVPp, and our method ASVP on **CIFAR-10** using (a) Margin, (b) BADGE, (c) Confidence, (d) Coreset and (e) ActiveFT (al) sampling.

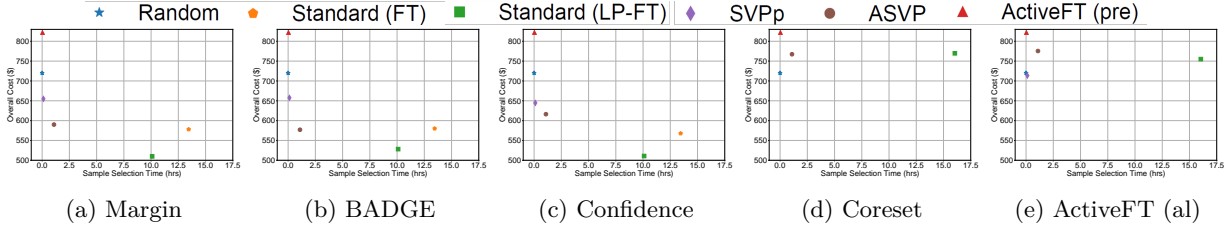

Figure 25: Computation efficiency and overall cost comparison of the standard active learning method, SVPp, and our method ASVP on **CIFAR-100** using (a) Margin, (b) BADGE and (c) Confidence sampling.

performance improvement diminishes, indicating a limit to the benefits of increasing the number of active learning iterations.

Additionally, the accuracy of SVPp (final model is fine-tuned) with more active learning iterations remains lower than that of the standard active learning pipeline. However, when using LP-FT for final model training

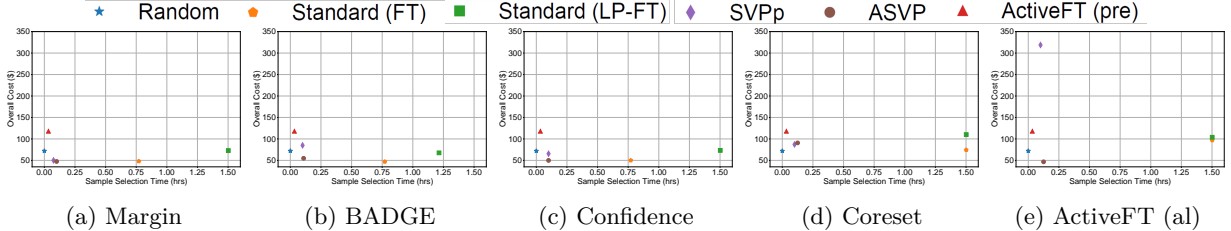

Figure 26: Computation efficiency and overall cost comparison of the standard active learning method, SVPp, and our method ASVP on **Pets** using (a) Margin, (b) BADGE and (c) Confidence sampling.

and increasing the iteration count to 45 or more, ASVP surpassed the standard active learning method in terms of accuracy.

Table 12: The influence of the number of active learning iterations on average sample savings ratios and sampling time on **ImageNet**.

| # AL iterations | Training Method | | Sampling Time (hrs) |
|---|---|---|---|
| | FT | LP-FT | |
| 9 | 16.64±2.06 | 30.22±1.85 | 1.1 |
| 45 | 17.66±1.37 | 31.20±0.40 | 1.5 |
| 180 | 19.43±0.86 | 32.00±0.46 | 3.7 |
| 900 | 18.21±0.67 | 31.64±0.24 | 15.8 |

# G  Replacement Experiment on CIFAR-10

We conducted replacement experiments on CIFAR-10 to demonstrate the impact of replacing missing samples from regions A1, A2, B1, and B2 on SVPp active learning performance. As shown in Figure 27, similar to the ImageNet replacement experiments, replacing samples from regions A1 and B1 results in little change in active learning performance, whereas replacing samples from regions A2 and B2 improves performance.

We also used Dataset Maps to illustrate the difficulty distribution of samples from different regions. The Dataset Map calculates the average and standard deviation of the model's confidence for the correct class (need true labels) using training dynamics from the entire dataset. Due to the simplicity of CIFAR-10, the Dataset Map shows very high mean confidence for almost all samples. Therefore, the difficulty based on mean confidence, as used in Section 4.1, would categorize nearly all samples as "easy".

Instead, we classified sample difficulty based on the standard deviation of the model's confidence in the correct class. Specifically, we defined the difficulty levels as follows: very hard (std. $>= 0.15$), hard (std. $>= 0.1$ and std. $< 0.15$), medium (std. $>= 0.05$ and std. $< 0.10$), and easy (std. $< 0.05$). The resulting distribution of different difficulty levels across regions is shown in Figure 28, where the proportion of each difficulty level within the entire dataset is displayed in the last column.

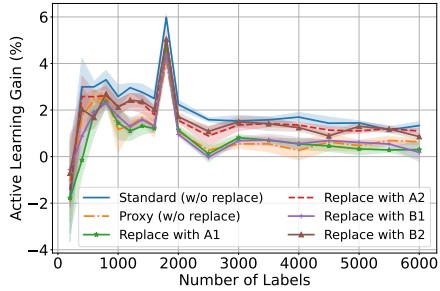

Figure 27: The impact of different regions on SVPp active learning performance. Replacing samples selected by the proxy model with those from regions A1, A2, B1, B2. The active learning gain refers to the difference in accuracy between active learning and the random baseline on CIFAR-10.

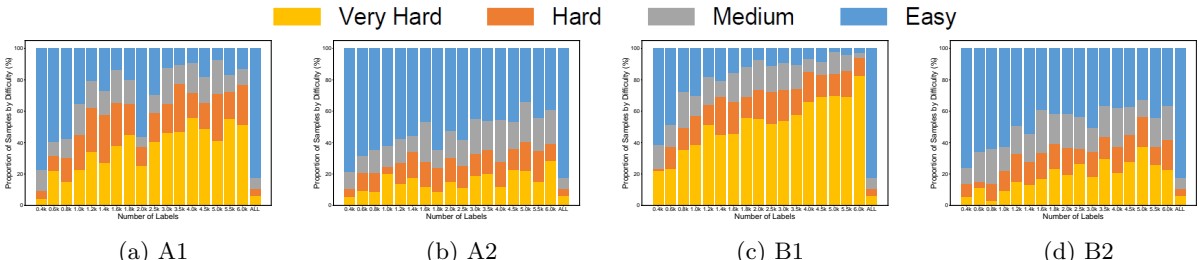

Figure 28: Difficulty distribution of samples from different regions on CIFAR-10.

