# OpenReview forum: "Feature Alignment: Rethinking Efficient Active Learning via Proxy in the Context of Pre-trained Models"
_TMLR — Accepted by TMLR_

### Review · Reviewer_ytc5 · 2024-07-26

**Summary Of Contributions:**

This paper proposes and investigates a more nuanced approach to fine-tuning large pre-trained models.  Fine-tuning models helps performance, but can be expensive on large models.  Actively selecting training points for fine tuning helps, but takes time to select the best points.  This work shows that sometimes the pre-trained model can be leveraged to actively select points quickly, and other times its representation of the margin is off, and one needs to iteratively update the model to get a better sense of margin, and get improved active learning.

The paper explains the ideas fairly well, and provides a reasonable set of experiments.  It shows the gains are modest, but still might be enough to make a difference.  Regardless of the degree of benefit, it looks like there may be some, and the overall study seems of interest to the ML community.

**Audience:**

Yes

**Claims And Evidence:**

Yes

**Requested Changes:**

Please address Weakness 2

**Strengths And Weaknesses:**

Strengths:
 1. The paper tackles an important problem, and provides a nuanced take.

 2. The writing style mixes explanation for design choices well with experiments.

 3. The paper seems to provide an honest take of pros and cons of the techniques


Weaknesses:
 1. The improvements of the proposed method seem small, and sporadic.

 2. The distinction between samples of type A vs B (and A1 vs. B1 and A2 vs B2) was confusing.  This seems to be whether the fine-tuned model has a positive or negative sign?  Or maybe was correct or incorrect (I am guessing from Figure 4).  This should be more clear in text.
  Moreover, if it is based on signs, then how does this make sense -- the sign choice should be semantic, and the algorithm should run the same if the signs are flipped?  Did you consider this (if it has to do with signs)?

---

> ### Author Response · Authors · 2024-08-25
> **Response**
>
> We would like to thank the reviewer's thoughtful comments, which we believe will strengthen the manuscript.
>
> > Weakness 1 The improvements of the proposed method seem small, and sporadic.
>
> As you pointed out, the improvement in total cost is modest. However, it is worth noting that the goal of this paper is to balance total cost and sample selection time rather than solely reducing total cost. To illustrate this, we present both the total cost and active learning sample selection time in Figure 13. We believe that enhancing the computational efficiency of active learning is valuable (it drives the research on efficient active learning [R1, R2]). For instance, improving the efficiency of AL can help address the issue of sample diversity that arises from batch selection. High efficiency allows us to select fewer samples per AL iteration without incurring significant computational costs, which can improve AL performance, as shown in Appendix Table 12. Additionally, it opens up possibilities for designing and applying algorithms that require extensive model retraining [R3].
>
>
> [R1] Coleman, Cody, et al. "Selection via Proxy: Efficient Data Selection for Deep Learning." International Conference on Learning Representations.
>
> [R2] Xie, Yichen, et al. "Towards free data selection with general-purpose models." Advances in Neural Information Processing Systems 36 (2024).
>
> [R3] Yang, Yazhou, and Marco Loog. "Single shot active learning using pseudo annotators." Pattern Recognition 89 (2019): 22-31.
>
>
> > Weakness 2 (1) The distinction between samples of type A vs B (and A1 vs. B1 and A2 vs B2) was confusing...This should be more clear in text.
>
> Thank you for your suggestion, we will update the following definition of region A1, A2, B1, and B2 in the subsequent revised version.
>
> Let $f(\cdot)$ represent the backbone and $h(\cdot)$ represent the predictor. The entire neural network model is $g = h(f(\cdot))$. The model fine-tuned on the labeled set $L = \\{ (x_i, y_i) \\}\_{i=1}^{|L|}$ is denoted as $g_{L} = h_{L}(f_{L}(\cdot))$. In standard active learning, $g_{L}(\cdot)$ is trained at each active learning iteration, and the active learning strategy $AS(\cdot)$ selects new samples from unlabeled set $U$ based on $g_{L}$. We use $AS(g_{L}(x_{i})) \text{ is True}$ to indicate that the sample $x_i$ is selected by the active learning strategy based on the fine-tuned model $g_{L}$. Let $h_{L_{p}}(\cdot)$ denote the proxy model trained on the labeled set $L_{p} = \\{(f_{p}(x_i), y_i) \mid x_i \in L\\}$ based on pre-computed features, where $f_{p}(\cdot)$ represents the backbone used to generate the pre-computed features. $AS(h_{L_{p}}(f_{p}(x_{i}))) \text{ is True}$ indicates that the sample $x_i$ is selected by the active learning strategy based on the proxy model $h_{L_{p}}(\cdot)$.
>
> Regions A and B are defined based on the conditions $A(g_{L}(\cdot)) \text{ is True}$ and $A(h_{L_{p}}(\cdot)) \text{ is False}$. In this paper, samples from these regions are further divided based on the correctness of predictions by $g_{L}(\cdot)$ and $h_{L_{p}}(\cdot)$ into subregions A1, A2, B1, and B2. Specifically:
>
> $A1 = \\{x' \mid x' \in U \text{ and } g_{L}(x') = y' \text{ and } h_{L_{p}}(f_{p}(x')) \neq y'\\}$
>
> $A2 = \\{x' \mid x' \in U \text{ and } g_{L}(x') = y' \text{ and } h_{L_{p}}(f_{p}(x')) = y'\\}$
>
> $B1 = \\{x' \mid x' \in U \text{ and } g_{L}(x') \neq y' \text{ and } h_{L_{p}}(f_{p}(x')) \neq y'\\}$
>
> $B2 = \\{x' \mid x' \in U \text{ and } g_{L}(x') \neq y' \text{ and } h_{L_{p}}(f_{p}(x')) = y'\\}$
>
> > Moreover, if it is based on signs, then how does this make sense -- the sign choice should be semantic, and the algorithm should run the same if the signs are flipped? Did you consider this (if it has to do with signs)?
>
> The division of these regions is used to illustrate that not all differences between proxy model-based active learning (AL) and fine-tuned model-based AL lead to performance disparities in AL (empirical evidence: Figure 5). To improve proxy model-based AL, we only need to focus on the regions that cause performance differences (i.e., A2, B2). As demonstrated in Section 4.2, using an appropriate training method like LP-FT can enhance the performance of proxy model-based AL (Figure 8). Thus, our algorithm does not require identifying these regions but rather focuses on selecting a suitable final model training method to bring proxy model-based AL performance closer to that of fine-tuned model-based AL. Section 4.3 discusses that as the total number of labeled samples increases, LP-FT may not fully bridge the gap between proxy model-based AL and fine-tuned model-based AL. To address this, we identify this point and update the pre-computed features to improve proxy model-based AL. Therefore, the two main operations of our ASVP method (selecting the appropriate training method and identifying the timing for updating pre-computed features) do not require identifying these regions.

---

> > ### Comment · Reviewer_ytc5 · 2024-08-26
> >
> > OK, thanks for the reply and also your detailed response to other reviewers.  I understand Fig 4 better now.  but I think the paper should label that [green-check] means the model predicts correctly and [red-x] means the model predicts incorrectly (as opposed to +/- labels).

---

> > > ### Author Response · Authors · 2024-08-27
> > >
> > > Thank you for your suggestion. We have updated Figure 4 in the revised version to include a legend for the [green-check] and [red-x].

---

### Review · Reviewer_MQfq · 2024-07-30

**Summary Of Contributions:**

The paper introduces a novel approach to proxy-based active learning that balances the trade-offs computational cost savings and model performance optimization. It shows that (1) only some types of sample selection result in the model's  performance degradation, and (2) suitable training methods can mitigate the decline of active learning performance. The proposed novel method, Aligned Selection Via Proxy (ASVP), improves proxy-based active learning performance by updating pre-computed features and selecting a proper training method.

**Audience:**

Yes

**Claims And Evidence:**

Yes

**Requested Changes:**

1. Fig 1 has pedagogical value, but the paper would greatly benefit from using at least one real-world motivating example in which SOTA is reached with realistic (a) annotation time/cost, and (b) model re-training time/cost. In many practical applications, getting an example's label is not instantaneous (as in CIFAR or ImageNet), but it may take significant time (eg, 3-7 days for 2K examples). In such scenarios, the time required to re-train the model may be dominated by the examples' labeling, so only the cost of retraining is a factor. Ideally, you should use a version of Fig 1 for such a realistic scenario.

2.  The example in Fig 2 seems to be poorly chosen, as the savings of Standard AL appear to be larger than for SVP AL (i.e., $854 = $961 - $107  vs   $805 = $808 - $3). Again, a variation of Fig 2 based on a real-world application would be greatly motivating.

3. Why did you change the setup between Figs 1 & 2 (i.e., 10x 200 vs 10x 10K)? Ideally, they should be two illustrations of the same experiment.

4. Section 4.1 is a good start, but should go deeper. Fig 6 is interesting, but it raises several questions:
(1) in itself, querying more difficult examples is NOT a bad thing, especially if the concept is "highly-learnable;" how many of these difficult ImageNet examples are part of the ~8% that SOTA still makes mistakes on? if you can identify-and-ignore such examples, you may have a (another) novel approach to AL.
(2) on a related note, could you please add the counter-part of Fig 6 for the other three datasets? are the findings holding on easier domains, like CIFAR-10, where nearly-perfect learning is possible? if yes, why is it bad to identify early these difficult examples? intuitively, it should actually help you converge faster.
(3) given that the overall proportion of very hard examples is comparable to the SOTA error rate on ImageNet, how would your zone A/B/C/D analysis change if you could simply identify-and-NeverQuery these difficult examples? Add a figue in which you compare ASVP vs this "oracle-fueled" strategy of NOT querying very hard examples? If "the oracle-base AL" is better, whichis a simpler strategy that focusing on A2/B2, it may be even worth creating an "online version/approximation" of Dataset Maps that would allow you to identify/ignore such examples.

5. Figs 7, 8, 9, 11: extend X axis until (near-)SOTA performance

6. In Table 1:
   - why is ASVP the top-performer on Cost but NOT on Avg Sample Saving Ratio?
   - how meaningful are the actual cost savings? again, a real-world example with real annotation costs would greatly help
   - is a cost saving ratio of 1% worth achieving/pursuing (ie, 74% vs 75%)? again, a real-world example would greatly help
   - how meaningfull are the ASVP cost savings, given that they are dominated by ImageNet, where the accuracy in Fig 11.a ends at 68% vs SOTA's 92%? Will these savings hold if you label another 200K/400K examples for ImageNet?

**Strengths And Weaknesses:**

The paper tackles an important practical issue: how to frugally improve the fine-tuning of a model. The proposed approach appears to be novel, original, sound, and potentially impactful. It also shed light on the root causes of the limitations of existing approaches.

The paper's main limitation is that it only analyses active learning in its early stages (that is, on the first N examples, as seen, for instance, in Fig 11) when the model's performance is still far from SOTA. For all 4 datasets in Fig 11, SOTA is far superior to the performance of AL, which raises three questions:
(1) how many more examples are required to reach SOTA?
(2) why should we care which approach is better at reaching a (sub-)mediocre performance?
(3) most importantly, will the paper's insights & findings hold once enough examples are labeled to reach SOTA(-adjacent) performance? This is closely related to the seminal findings of [Banko & Brill, 2001] on what area of a learning curve does performance really matter.

[Banko & Brill, 2001] Banko, Michele, and Eric Brill. "Scaling to very very large corpora for natural language disambiguation." Proceedings of the 39th annual meeting of the Association for Computational Linguistics. 2001.

---

> ### Author Response · Authors · 2024-08-25
> **Response to points 1~3**
>
> Thank you for your thoughtful and thorough review. Your constructive comments and suggestions have been incredibly helpful in improving the quality of our paper！
>
> > 1. Fig 1 has pedagogical value, but the paper would greatly benefit from using at least one real-world motivating example in which SOTA is reached with realistic (a) annotation time/cost, and (b) model re-training time/cost. In many practical applications, getting an example's label is not instantaneous (as in CIFAR or ImageNet), but it may take significant time (eg, 3-7 days for 2K examples). In such scenarios, the time required to re-train the model may be dominated by the examples' labeling, so only the cost of retraining is a factor. Ideally, you should use a version of Fig 1 for such a realistic scenario.
>
> (a) Thank you for your suggestion. We have extended the experiments on ImageNet and CIFAR-10 to approach the state-of-the-art performance. We will update figure 1 \& 2 in the subsequent revised versions. We have included the extended accuracy curves in Figure 11, and the updated cost table is provided in our response to Request Change 6(4).
>
> (b) Thank you for your insightful comment. While manual labeling indeed consumes significant time, it's important to note that in practical applications, manual labeling is often carried out through crowdsourcing. The cost of manual labeling is primarily related to the total number of labels required and the complexity of the labeling task, rather than the number of labelers working simultaneously. Consequently, increasing the number of labelers can significantly reduce the time required for manual labeling.
>
> Additionally, the computational efficiency of active learning is crucial, which is why efficient active learning methods have been developed [R1, R2]. Improving the computational efficiency of active learning can facilitate further research possibilities. For instance, enhancing the computational efficiency can address the performance degradation issue associated with batch selection in standard active learning. Since training deep neural networks is relatively costly, existing active learning methods often select samples in batches, which may result in insufficient diversity in the selected samples. Efficient active learning methods, with lower model retraining costs, can enable smaller batch sizes (i.e., increase the number of active learning iterations), thereby alleviating the diversity issue inherent in batch selection. This is demonstrated in Appendix Table 12.
>
> Furthermore, improving the computational efficiency of active learning also makes it feasible to apply algorithms that require extensive retraining, such as [R3].
>
>
> [R1] Coleman, Cody, et al. "Selection via Proxy: Efficient Data Selection for Deep Learning." International Conference on Learning Representations.
>
> [R2] Xie, Yichen, et al. "Towards free data selection with general-purpose models." Advances in Neural Information Processing Systems 36 (2024).
>
> [R3] Yang, Yazhou, and Marco Loog. "Single shot active learning using pseudo annotators." Pattern Recognition 89 (2019): 22-31.
>
> > 2. The example in Fig 2 seems to be poorly chosen, as the savings of Standard AL appear to be larger than for SVP AL. Again, a variation of Fig 2 based on a real-world application would be greatly motivating.
>
> Thank you for pointing this out. The term "SVP AL" refers to existing efficient active learning algorithms. We use this figure to demonstrate that while the existing efficient AL algorithm (SVPp in experiment section) reduces AL training time (and cost), it also weakens active learning performance, resulting in total costs (labeling + training) that are higher than those for standard active learning.
>
> We would like to clarify the savings comparison under this experimental setup for Standard AL, SVP AL, and our proposed ASVP method as follows:  Stanrda AL (FT): 854 = 961 - 107, Stanrda AL (LP-FT): 1139 = 1247 - 107, SVP AL: 805 = 808-3, ASVP (ours): 1219=1222-3.
>
> Additionally, we will address Requested Changes 2 and 3 by updating Figures 1 and 2 to use the same dataset and by including the ASVP results in Figure 2.
>
> > 3. Why did you change the setup between Figs 1 & 2 (i.e., 10x 200 vs 10x 10K)? Ideally, they should be two illustrations of the same experiment.
>
> Since Figure 1 presents experiments conducted on CIFAR-10, which requires fewer labeled samples to achieve satisfactory accuracy, and Figure 2 presents experiments on ImageNet, which requires a larger number of labeled samples to reach a decent accuracy. So We used different sampling sizes per round. Thank you for your suggestion. We will update the figures in a subsequent version to ensure they are based on the same dataset.

---

> ### Author Response · Authors · 2024-08-25
> **Response to points 4.(1) & (2)**
>
> > Section 4.1 is a good start, but should go deeper. Fig 6 is interesting, but it raises several questions: (1) in itself, querying more difficult examples is NOT a bad thing, especially if the concept is "highly-learnable;" how many of these difficult ImageNet examples are part of the ~8% that SOTA still makes mistakes on? if you can identify-and-ignore such examples, you may have a (another) novel approach to AL.
>
> (a) We will address the question regarding querying difficult samples in our response to Request 4.2.
>
> (b) In Figure 6, we use Dataset Maps to illustrate the proportions of samples with varying difficulty levels within different regions. The training dynamics used in the Dataset Maps are derived from training a ResNet-50 model on the whole ImageNet dataset. Therefore, the "unlearnable" samples  (i.e., samples on which the model still makes mistakes after training on the entire dataset) for this model correspond closely to the "very hard" category in Figure 6. Specifically, approximately 75\% and 33\% of the samples in the "very hard" and "hard" categories in Figure 6, respectively, are unlearnable samples. Furthermore, in our experiments using a ResNet-50 model (pre-trained with BYOL-EMAN) and then trained on all ImageNet labels, the model still makes mistakes on about 21\% of the samples ("unlearnable" samples).
>
> (c) Yes, identifying these samples during the active learning could potentially lead to a new active learning method. However, the Dataset Map and the regions A1, A2, B1, and B2 in our paper rely on the true labels of the unlabeled samples within the active learning pool. Therefore, at this stage, we can only use these tools for analysis and are unable to apply them to develop a new active learning strategy.
>
> > 4. (2) on a related note, could you please add the counter-part of Fig 6 for the other three datasets? are the findings holding on easier domains, like CIFAR-10, where nearly-perfect learning is possible? if yes, why is it bad to identify early these difficult examples? intuitively, it should actually help you converge faster.
>
> Thank you for your insightful suggestions. We have added corresponding results for CIFAR-10 in Appendix G, including (1) comparing AL performance after replacing samples in regions A1, A2, B1, and B2 and (2) showing the proportion of samples with different difficulty levels in each region. The experimental results support our findings: regions A1 and B1 contain more difficult samples compared to regions A2 and B2, and replacing samples selected by the proxy model with those from A1 and B1 does not significantly improve AL performance based on the proxy model.
>
> We believe this result is due to two main factors. First, as you mentioned, for challenging datasets (with a certain proportion of unlearnable samples), replacing samples with unlearnable ones cannot increase AL performance. Second, increasing the proportion of difficult samples reduces the proportion of easy samples. This leads to a noticeable improvement in the model's prediction accuracy on difficult samples within the active learning pool and test set, but a limited improvement on easy samples. Since we aim to maximize the overall prediction accuracy across all sample difficulties, increasing the difficulty of selected samples does not necessarily lead to a monotonic increase in model performance.
>
> To illustrate this, consider the replacement experiments on the CIFAR-10 dataset, which are similar to those in Figure 5, where samples from regions B1 (with a high proportion of difficult samples) and B2 (with a low proportion of difficult samples) were used to replace those selected by the proxy model. As shown in the following table, after replacing with samples from B1, the model shows a greater improvement in prediction accuracy for "very hard" and "hard" samples within the active learning pool, but less improvement for "medium" and "easy" samples. As a result, replacing with B2 (lower difficulty) leads to a greater overall improvement in the model's prediction accuracy across the entire active learning pool compared to replacing with B1 (higher difficulty).
>
> **Comparison of model accuracy on active learning pool (for different difficulty samples and for the whole active learning pool) after replacing samples of different difficulty levels, with the number of 'Very Hard,' 'Hard,' 'Medium,' and 'Easy' samples in the entire active learning pool being 1364, 1573, 3658, and 43405, respectively.**
> |                            | Very Hard | Hard  | Medium | Easy  | All   |
> |----------------------------|-----------|-------|--------|-------|-------|
> | Replace with B1            | 42.4%     | 63.8% | 78.4%  | 97.3% | 93.3% |
> | Replace with B2            | 37.1%     | 60.3% | 79.8%  | 98.1% | 93.9% |

---

> > ### Author Response · Authors · 2024-08-25
> > **Response to points 4.(3) & 5**
> >
> > > 4. (3) given that the overall proportion of very hard examples is comparable to the SOTA error rate on ImageNet, how would your zone A/B/C/D analysis change if you could simply identify-and-NeverQuery these difficult examples? Add a figue in which you compare ASVP vs this "oracle-fueled" strategy of NOT querying very hard examples? If "the oracle-base AL" is better, whichis a simpler strategy that focusing on A2/B2, it may be even worth creating an "online version/approximation" of Dataset Maps that would allow you to identify/ignore such examples.
> >
> > Thank you for your suggestion. We can only use Dataset Maps as a post-analysis tool, rather than applying it for sample selection in active learning. This is because, during active learning, the labels of the queried samples are unknown. In contrast, Dataset Maps require training a model on the entire dataset and calculating the mean and standard deviation of the model's predictions for the correct class to assess sample learnability. To clarify this for readers, we have revised the description of Dataset Maps in the paper as follows, "Dataset Maps depict sample learnability by analyzing training dynamics from models trained on the entire dataset, using metrics such as mean confidence in the correct class and prediction accuracy across training epochs to quantify difficulty."
> >
> > >5. Figs 7, 8, 9, 11: extend X axis until (near-)SOTA performance
> >
> > Thank you for your suggestions. We will extend the results for Figures 7, 8, and 9 in a subsequent version. It is worth noting that the extension of Figures 7, 8, and 9 will not affect the main conclusions of our paper. These figures illustrate the motivation behind the ASVP method. Figures 7 and 8 support the idea that, before updating pre-computed features, the ASVP method can reduce the performance gap between efficient active learning algorithms (based on the proxy model) and standard active learning algorithms (based on the fine-tuned model) by using LP-FT training. Figure 9 illustrates that as the number of labeled samples increases, the quality of samples selected by the proxy model deteriorates over time, which necessitates updating the pre-computed features.
> >
> > For the ImageNet experiments in Figures 7, 8, and 9, the ASVP method updates pre-computed features at approximately 120k labeled samples (near the endpoint of these figures). After this update, ASVP no longer uses LP-FT, so whether the trends observed in Figures 7 and 8 continue does not impact the conclusions of our paper.
> >
> > Additionally, we will extend Figure 5 in subsequent versions. After the extension (we are running the rest of the replacement experiments), the results of replacing with B1 and B2 still support the current conclusions of this paper: replacing with samples from region B1 does not significantly improve the active learning performance based on the proxy model, whereas replacing with samples from region B2 does lead to improvement. The results are shown in the table below.
> >
> > **Replacement experiment on ImageNet (extension of figure 5).**
> > | \# labels          | 150k   | 200k   | 250k   | 300k   | 350k   | 400k   |
> > |--------------------|--------|--------|--------|--------|--------|--------|
> > | Proxy (w/o replace) | 70.59% | 71.87% | 72.80% | 73.60% | 74.07% | 74.61% |
> > | Replace with B1    | 69.94% | 71.50% | 72.56% | 73.22% | 73.88% | 74.48% |
> > | Replace with B2    | 70.81% | 72.26% | 73.20% | 73.88% | 74.46% | 74.98% |
> >
> > Regarding Figure 11(d), the endpoint performance of 91.3\% is already close to the state-of-the-art performance of 91.6\%. In the revised version, we have updated the experiments for ImageNet ( figure 11(a) ) and CIFAR-10 ( figure 11(b) ). For ImageNet, we extended to 400k samples, achieving an accuracy of approximately 75\%, which is close to the state-of-the-art accuracy of about 77\% (using ResNet-50 with BYOL-EMAN pre-training). For CIFAR-10, we extended to 6000 samples, achieving an accuracy of approximately 96\%, which is close to the state-of-the-art accuracy of about 97.5\%.

---

> > > ### Author Response · Authors · 2024-08-25
> > > **Response to points 6.(1)-(3)**
> > >
> > > > 6. In Table 1: (1) why is ASVP the top-performer on Cost but NOT on Avg Sample Saving Ratio?
> > >
> > > There are two reasons for this. First, the cost reported in Table 1 includes both labeling cost and model training cost. The training cost required for standard active learning significantly exceeds that of SVPp and ASVP, which may result in ASVP having a lower overall cost but requiring slightly more labeling cost. Second, Table 1 reports the average sample saving ratio, which reflects the proportion of samples saved compared to random sampling (fine-tuning) throughout the active learning process (e.g. average savings from \#20k to \#100k at ImageNet experiments). In contrast, the cost reported refers to the total cost (labeling cost + model training cost) required for the active learning method to achieve the accuracy of random sampling at the experiment’s endpoint (e.g. AL reaches the accuracy of random sampling at \#100k for the ImageNet experiment).
> > >
> > > We chose this setup because active learning may terminate at varying amounts of labeled data. In practical application, the accuracy trained on the entire dataset is unknown (for an unknown dataset). Therefore, active learning may be terminated when the model reaches a satisfactory accuracy, or when the labeling and model training costs hit the budget limit. Thus, both the performance throughout the active learning phase and the endpoint performance are relevant in real-world scenarios. To reflect performance across the entire active learning phase, we use the average sample saving ratio, while the total cost reflects performance at the endpoint.
> > >
> > > > 6. In Table 1: (2) how meaningful are the actual cost savings? again, a real-world example with real annotation costs would greatly help
> > >
> > > Thank you for your suggestion. We have extended the ImageNet experiment to 400k labeled samples, where the AL accuracy at the stopping point is approximately 75\% (compared to around 77\% when using the entire dataset with ResNet-50 pre-trained by BYOL-EMAN). For the CIFAR-10 experiment, we extended it to 6000 labeled samples, where the AL accuracy at the stopping point is about 96\% (compared to around 97.5\% when using the entire dataset). In Table 1, the cost includes both labeling and training costs. The results of the extension experiment are detailed in the table below (in our response to Request 6(4)).
> > >
> > > Additionally, as mentioned in our response to Request 6(1), we believe that both the performance throughout the active learning phase and the performance at the stopping point, when a reasonable accuracy is reached, are meaningful for practical applications. Therefore, we use the cost and cost-saving ratio in Table 1 to reflect performance at the stopping point and the average sample-saving ratio to reflect performance across the entire active learning range. We also recognize that, in addition to the total cost, the time taken for sample selection in active learning is of practical significance (as noted in our response to Request Change 1). Thus, we use Figure 13 to present the results for time spent and total cost.
> > >
> > > > 6. In Table 1: (3) is a cost saving ratio of 1% worth achieving/pursuing (ie, 74% vs 75%)? again, a real-world example would greatly help
> > >
> > > The goal of this paper is to balance total cost and sample selection time rather than solely reducing total cost. We believe that reducing the time required for active learning sample selection is valuable, as discussed in our response to Requested Change 1. This is the motivation of existing efficient active learning methods [R1, R2]. However, the problem we aim to address is that existing efficient active learning methods tend to reduce sample selection time (training cost) at the expense of active learning performance, compared to selecting samples using a fine-tuned model. This trade-off forces practitioners to choose between saving time or reducing overall cost. Our method, on the other hand, improves active learning performance while maintaining a short sample selection time, resulting in a total cost that is close to or slightly better than standard active learning (using a fine-tuned model). This means that users can benefit from both the time efficiency of existing efficient AL methods and the high performance of standard active learning.

---

> > > > ### Author Response · Authors · 2024-08-25
> > > > **Response to point 6.(4)**
> > > >
> > > > >6. In Table 1: (4) how meaningfull are the ASVP cost savings, given that they are dominated by ImageNet, where the accuracy in Fig 11.a ends at 68% vs SOTA's 92%? Will these savings hold if you label another 200K/400K examples for ImageNet?
> > > >
> > > > (a) To clarify, the cost saving ratio reported in Table 1 is averaged across datasets, meaning that the high total cost of ImageNet does not dominate the overall ratio. For example, for the standard (FT) margin sampling in Table 1:
> > > >
> > > > For ImageNet, with a total cost of 2739, the cost saving ratio is 2739 / 3600 = 76.1\% (where 3600 is the labeling cost for random sampling).
> > > > For CIFAR-10, with a total cost of 44, the cost saving ratio is 44 / 72 = 61.1\%.
> > > > For CIFAR-100, with a total cost of 207, the cost saving ratio is 207 / 216 = 95.8\%.
> > > > For Pets, with a total cost of 48, the cost saving ratio is 48 / 72 = 66.7\%.
> > > > The average cost saving ratio reported in Table 1 is thus (76.1\% + 61.1\% + 95.8\% + 66.7\%) / 4 = 75\%.
> > > >
> > > > (b) For the ImageNet experiments, the ResNet-50 model (pre-trained with BYOL-EMAN) achieves an accuracy of about 77\% when trained on the entire dataset. In this paper, the reported stopping point accuracy is about 68\%. Following your suggestion, we extended the ImageNet experiments to a total of 400k labeled samples (using margin sampling). The results are shown in the table below， where Eqv. # labels of random sampling refers to the number of labeled samples required by random sampling to achieve the accuracy shown in the table
> > > >
> > > > **Comparison of total costs with margin sampling, with values in () indicating cost saving ratio (Cost of AL / Cost of Random Sampling).**
> > > >
> > > > | Dataset   | Accuracy | Eqv. \# labels of random sampling | Random (FT) | Margin Standard (FT) | Margin SVPp (FT) | Margin ASVP (LP-FT/FT) |
> > > > |-----------|----------|----------------------------------|-------------|-----------------------|------------------|------------------------|
> > > > | ImageNet  | 72.9%    | 400000                           | \$14400 (100%) | \$8865 (61.56%)       | \$9131 (63.41%)  | \$8728 (60.61%)       |
> > > > | ImageNet  | 75.0%    | 680000                           | \$24480 (100%) | \$14963 (61.12%)      | \$24623 (64.20%) | \$14907 (60.89%)      |
> > > > | CIFAR-10  | 94.7%    | 6000                             | \$216 (100%)  | \$132 (61.11%)        | \$169 (78.24%)   | \$128 (59.26%)        |

---

> > > > > ### Comment · Reviewer_MQfq · 2024-08-26
> > > > > **Thank you for the detailed answers**
> > > > >
> > > > > Thank you very much for your detailed answers, which are extremely helpful.
> > > > >
> > > > > I have one clarification of my own, as my comment was ambiguous: when I said "SOTA performance," I was referring to the base learner that, when trained on all data, achieves the best known performance. I fully understand that you extended the nmb of examples to "close-to-SOTA for ResNet-50" (ie, 75% vs 77%), which, IMHO, is perfectly fine for thre purpose of this review. However, what I should have said is "with the best base learner, ImageNet's SOTA is 93.6%, vs the value that you had in the original Fig 11. I am also wondering how difficult would it be to use/re-implement as base learner one of the 19 approaches that gets to 90+% in https://paperswithcode.com/sota/image-classification-on-imagenet

---

> > > > > > ### Author Response · Authors · 2024-08-28
> > > > > >
> > > > > > Thank you for your insightful suggestion regarding the use of SOTA models with 90+% accuracy as the base learner. We appreciate the value of these models; however, most of these methods require extremely large models with billions of parameters (e.g., CoCa with 1 billion, BASIC-L with 2.4 billion). Implementing such large models in active learning experiments, over multiple iterations on a large dataset like ImageNet, would require substantial GPU resources, which, unfortunately, is beyond our current capacity.
> > > > > >
> > > > > > We did consider using models closer in scale to ResNet-50, such as CoCa-Base (86 million parameters). However, the performance improvement over our current setup is quite limited. For instance, using margin sampling, CoCa-base reaches around 75.5% accuracy with 400k samples (Figure 4 (a) [R4]), which is very close to the 75% achieved by ResNet-50 in our experiments.
> > > > > >
> > > > > > [R4] Zhang, Jifan, et al. "LabelBench: A Comprehensive Framework for Benchmarking Adaptive Label-Efficient Learning." Journal of Data-centric Machine Learning Research (2024).

---

> > > > > > > ### Comment · Reviewer_MQfq · 2024-08-29
> > > > > > > **On large base learners**
> > > > > > >
> > > > > > > I agree with you position. However, as part of the discussion section, you should cover all these topics. For each application domain, it should be clear what one can expect from your approach, and what can be expected from larger base models. My point is: get in front of the problem, rather than have a reader/reviewer "work their way through it." Such insights may be, in some way, (almost) as valuable as a paper's technical contribution.

---

> > > > > > > > ### Author Response · Authors · 2024-09-03
> > > > > > > >
> > > > > > > > Thank you for your suggestion. We have included a limitations section in the revised version of the paper.

---

### Review · Reviewer_KTcw · 2024-08-11

**Summary Of Contributions:**

The paper's contributions are three folded:
1. It empirically analyzes the sample selection differences that contribute to the performance drop when employing Selection
via Proxy based on pre-trained features  (SVPp).
2. It proposes a novel efficient active learning approach, Aligned Selection via Proxy (ASVP), which improves the performance of efficient active learning while incurring a marginal amount of computational time.
3. It introduces an evaluation metric for efficient active learning, the Sample Saving Ratio, which directly quantifies the savings in annotation achieved by employing active learning strategies compared to the random baseline.

**Audience:**

Yes

**Broader Impact Concerns:**

No specific impact concerns.

**Claims And Evidence:**

No

**Requested Changes:**

1. I suggest the authors restructure the paper's organization. The current one reads a bit confusing as the major contributions are not emphasized. The observations can be shortened and some of them are experimental results (should be moved to experiment sections).

2. emphasize why the observations and the proposed evaluation metric is novel.

**Strengths And Weaknesses:**

Strengths:
1. The paper addresses an important problem in learning.
2. The proposed method upgrades the second stage in SVPp where SVPp's data selection uses only a simple classifier to select data which is hard to sample a good distribution of examples. The methodology is sound.

Weaknesses:
1. Overestimation of contributions: the diversity of sample distributions are important in active learning, which is known in the field. also, the metric for evaluating active learning does not seem novel. It is equivalent to sample size @ certain accuracy which is a common way to evaluate learning methods.
2. The paper is a bit hard to follow and confusing.
3. Based on (1), I think the paper's contribution is quite limited and doesn't really worth publishing a long paper. Could be interesting workshop paper though.

---

> ### Author Response · Authors · 2024-08-25
>
> Thank you for your time in reviewing our paper and for your thoughtful feedback!
>
> > Requested Change 1 and Weakness 2 The paper is a bit hard to follow and confusing.
>
> Thank you for your suggestion. We plan to reorganize Section 4 in the future version. We intend to adopt a hypothesis-empirical evidence structure for clarity. Specifically, Section 4.1 will present the hypothesis that not all differences in sample selection between SVPp and standard AL contribute to the AL performance differences. The empirical evidence for this is shown in Figure 5. We will move the further analysis of the causes (currently in Figure 6) to the experimental section. Section 4.2 will focus on the hypothesis that the impact of SVPp missing samples in regions A2 and B2 can be mitigated by selecting an appropriate training method, with Figures 7 and 8 providing empirical evidence. The remaining discussion in Section 4.3 will be integrated into the methods section (Section 5), and the empirical evidence from Section 4.3 (currently in Figure 9) will be moved to the experimental section.
>
> > Requested Change 2 emphasize why the observations and the proposed evaluation metric is novel.
>
> **Observation**: Existing proxy model-based efficient active learning methods advocate for selecting samples that are as similar as possible to standard active learning (which relies on a fine-tuned model), using ranking correlation to assess the quality (consistency) of the samples selected by the proxy model [R1]. However, our observations in Section 4.1 (Figure 5) show that it is not necessary for the proxy model to select the exact same samples as the fine-tuned model. This is because some of the regions (A1, B1) selected by the fine-tuned model contain too many "hard-to-learn" samples (as shown in Section 4.1, Figure 6), and missing these samples does not negatively impact the performance of proxy model-based active learning. In other words, the observation in Section 4.1 shows that not all differences in sample selection between SVPp and standard AL contribute to the AL performance differences.
>
> Moreover, the samples that do affect the performance of proxy model-based active learning (those in regions A2 and B2) do not need to be identified explicitly. Instead, simply adopting an appropriate training method when training the final model (as shown in part (3) of Figure 3) can compensate for the impact of missing these samples on AL performance (as demonstrated in Figures 7 and 8). Based on these observations (and the discussion in Section 4.3), we propose a simple yet effective method, ASVP. Compared to the existing efficient active learning method SVPp, ASVP significantly improves AL performance while maintaining the efficiency of sample selection time when compared to standard active learning (Figure 13).
>
> [R1] Coleman, Cody, et al. "Selection via Proxy: Efficient Data Selection for Deep Learning." International Conference on Learning Representations.
>
> **Evaluation**:  To our knowledge, this paper is the first to recommend evaluating efficient active learning methods by considering total cost, which includes both training and labeling costs. We observed that existing efficient AL methods often reduce sample selection time (i.e., training cost) but at the expense of active learning performance. Although the drop in AL accuracy may seem minor (depending on the dataset and number of labeled samples), it can lead to higher total costs when calculating total cost (training and labeling, figure 2). This issue hasn't been explored in previous research.
>
> We believe this is critical for the practical application of efficient active learning since the primary goal of AL is to reduce labeling costs. If efficient AL increases total costs while only saving time, it could undermine its value. Therefore, we advocate for explicitly comparing total costs in evaluations. To make this easier, we calculate the equivalent number of labeled samples to quantify labeling cost savings, which can then be added to training costs to determine the total cost.
>
> >Weakness 1 and Weakness 3
>
> We are not sure which part you are referring to with "diversity of sample distribution". This paper primarily explores an issue that has been unexplored in existing efficient AL, i.e. balancing sample selection time and total cost (less time and less cost). In our response to Requested Change 2, we discuss the observations of this paper, why they are important, and why we advocate for reporting total cost in efficient active learning.
>
> Yes, the \#labels \@ accuracy plot shows the number of labels needed for an AL method to reach a certain accuracy, but it does not accurately convey the number. Additionally, this paper emphasizes reporting the total cost (labeling cost + training cost) for efficient AL. Including the equivalent number of labels in the total cost calculation helps readers accurately and conveniently understand the proportion of labeling cost in the total cost.

---

### Decision · Action_Editor_5EKG · 2024-10-10

**Recommendation:** Accept with minor revision

**Comment:**

The reviewers think that the paper is of interest to TMLR audience. Regarding the supporting evidence, the reviewers are mostly satisfied with the responses, but they still think the current revision needed more work. Specifically they requested the following changes in the final version:

- Revise Figure 2 (so that the training costs of SVP_p is clear)
- Add explanations in the paper on why experiments are only evaluated on smaller models rather than SoTA large models
- For Fig 11, add explanations on why only two of the four axis were extended, and what is the expected performance upper bound if we use the full dataset

@authors: as we enter the final revision stage, please feel free to follow up if any clarification on the requests are needed.

**Audience:**

Yes. This paper will be of interest to practitioners of deep active learning that aim at balancing computation efficiency with label efficiency.

**Claims And Evidence:**

Yes, mostly. See final comments below.

---

> ### Author Response · Authors · 2024-10-29
>
> Dear Editor,
>
> We greatly appreciate the valuable feedback provided by you and the reviewers. Based on your suggestions, we have made the following changes to the revised manuscript:
>
> 1. We have revised Figure 2 to make the training cost, labeling cost, and sampling time for Standard AL, SVPp, and ASVP (ours) clearer.
> 2. In Section 8 (Limitations), we included a discussion on the limitation of not conducting experiments on larger-scale models due to limited computational resources. This section has been highlighted in blue in the revised manuscript.
> 3. We have extended the remaining two datasets’ experiments to approach the expected performance upper bound and updated Figure 11 to reflect this upper bound. Correspondingly, we have also updated Table 1, Figure 12, Figure 13, and the related results in the appendix with the extended experimental results.
>
> Thank you again for the guidance on the revised revision.